# High temperature delays and low temperature accelerates evolution of a new protein phenotype

Jia Zheng [1,2,3,7] ✉, Ning Guo [1,2,3,7], Yuxiang Huang[1,2,3], Xiang Guo[1,2,3] & Andreas Wagner [4,5,6] ✉

Since the origin of life, temperatures on earth have fluctuated both on short and long time scales. How such changes affect the rate at which Darwinian evolution can bring forth new phenotypes remains unclear. On the one hand, high temperature may accelerate phenotypic evolution because it accelerates most biological processes. On the other hand, it may slow phenotypic evolution, because proteins are usually less stable at high temperatures and therefore less evolvable. Here, to test these hypotheses experimentally, we evolved a green fluorescent protein in *E. coli* towards the new phenotype of yellow fluorescence at different temperatures. Yellow fluorescence evolved most slowly at high temperature and most rapidly at low temperature, in contradiction to the first hypothesis. Using high-throughput population sequencing, protein engineering, and biochemical assays, we determined that this is due to the protein-destabilizing effect of neofunctionalizing mutations. Destabilization is highly detrimental at high temperature, where neofunctionalizing mutations cannot be tolerated. Their detrimental effects can be mitigated through excess stability at low temperature, leading to accelerated adaptive evolution. By modifying protein folding stability, temperature alters the accessibility of mutational paths towards high-fitness genotypes. Our observations have broad implications for our understanding of how temperature changes affect evolutionary adaptations and innovations.

Temperature influences all processes in the biosphere. On the smallest scale of biomolecules, it affects processes such as protein folding and catalytic efficiency[1–6]. On the intermediate scale of organisms, it affects the activity and metabolic rate of individuals[7–9]. On the largest scale of ecosystems, it affects biodiversity and productivity[1,10–13]. These large-scale effects are ultimately also caused by temperature's effects on the molecular scale[1,14–18].

Most modern proteins are derived from the last universal common ancestor of today's organisms, which existed more than 3.5 billion years ago[19,20]. Since this time, temperature on Earth has changed dramatically on both long- and short-time scales. We know that organisms can adapt to such changes by acquiring adaptive DNA mutations[5,14,15,21,22]. However, we still lack direct experimental evidence on how temperature affects the rate at which new phenotypic traits evolve. On the one hand, increasing temperature can accelerate both enzyme-catalyzed and spontaneous chemical reactions. It can also accelerate protein folding by increasing the rate at which molecules collide[23,24]. On the other hand, the very same collisions can cause

[1]Zhejiang Key Laboratory of Structural Biology, School of Life Sciences, Westlake University, Hangzhou, China. [2]Westlake Laboratory of Life Sciences and Biomedicine, Hangzhou, China. [3]Institute of Biology, Westlake Institute for Advanced Study, Hangzhou, China. [4]Department of Evolutionary Biology and Environmental Studies, University of Zurich, Zurich, Switzerland. [5]Swiss Institute of Bioinformatics, Lausanne, Switzerland. [6]The Santa Fe Institute, Santa Fe, USA. [7]These authors contributed equally: Jia Zheng, Ning Guo. ✉e-mail: zhengjia@westlake.edu.cn; andreas.wagner@ieu.uzh.ch

increasing temperatures to destabilize, denature, and inactivate proteins. These two opposing factors also entail conflicting selection pressures during the adaptive evolution of proteins. At elevated temperatures, proteins must evolve greater stability to reduce heat denaturation and maintain structural integrity[2,3,23]. At lower temperatures, proteins need to evolve greater flexibility to combat the slowing down of molecular motion[2,25–27]. These opposing demands make it difficult to predict whether temperature changes would facilitate or hinder the evolution of a new protein phenotype.

Comparative and empirical studies have focused on the effects of temperature on the evolution of physiological performance, such as biomass growth[15,28,29], but how temperature affects the evolution of new phenotypes has not been studied. Because increasing temperature accelerates most processes in the biosphere[1,13,30], it may also accelerate the evolution of new phenotypes. Alternatively, increasing temperature may impede phenotypic evolution, because high temperature can destabilize proteins and thus reduce protein evolvability[31–33]. We validated these conflicting hypotheses by experimentally evolving green fluorescent protein (GFP, a derivative of the green fluorescent protein of the jellyfish *Aequorea Victoria* (avGFP); see Fig. S1)[34] in *E. coli*. Because GFP is neither native to nor essential for *E.coli*, it interferes minimally with the *E.coli* proteome and physiology,

and is thus especially well-suited for this purpose. We asked how a change in temperature affects the ability of GFP to evolve the new color phenotype of yellow fluorescence (Fig. 1a). During each of five rounds ("generations") of mutation and selection, we mutagenized the *gfp* gene and selected cells for survival based on single-cell fluorescence phenotypes that are quantifiable through fluorescence-activated cell sorting (FACS) – another key benefit of using this protein. The starting ("ancestral") GFP we used is well adapted to the native 37 °C growth temperature of *E.coli*[34]. We evolved it both at this temperature, as well as at a higher and lower temperature. We also monitored its evolutionary dynamics via single-molecule real-time sequencing (SMRT), and identified the effect of key genetic changes through mutant engineering and biochemical assays. In this work, we show that low temperature rather than high temperature can promote phenotypic adaptation through neofunctionalizing mutations – mutations that bring forth a new protein phenotype.

## Results

### Elevated temperature delays and lowered temperature accelerates evolution of a new phenotype

We evolved GFP in populations of at least $2 \times 10^6$ *E. coli* cells towards yellow fluorescence at 25 °C (*L* for low temperature), 37 °C (*M* for

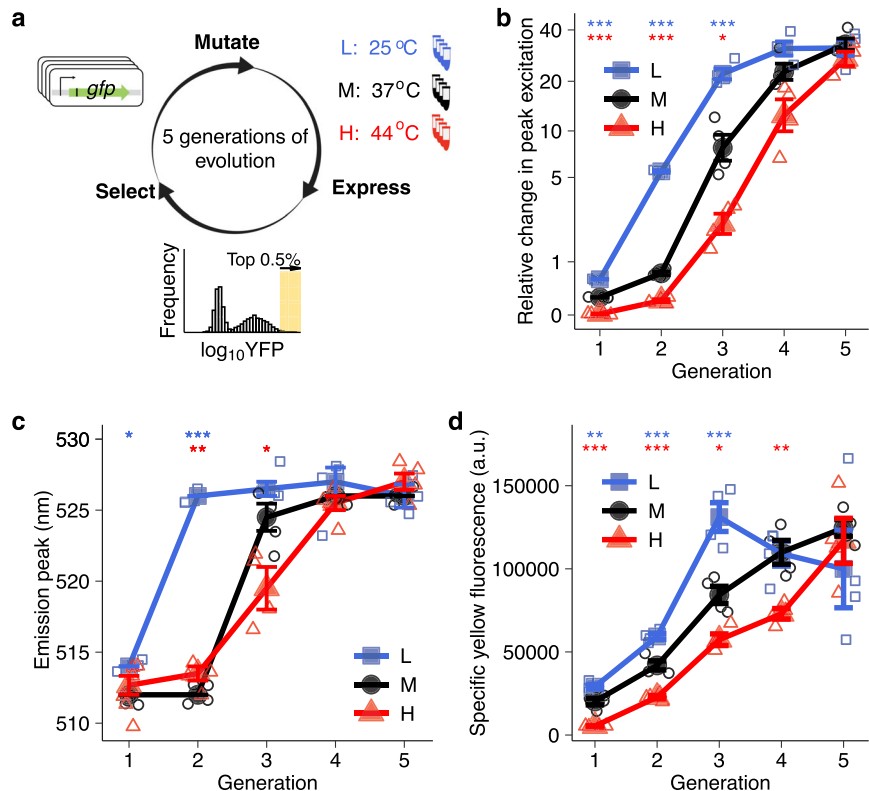

**Fig. 1 | Elevated temperature delays and lowered temperature accelerates the evolution of a new yellow fluorescence phenotype. a** Experimental design. We subjected four replicate populations of GFP to five generations of directed evolution at 25 °C (blue; populations *L*), 37 °C (black; populations *M*) or 44 °C (red; populations *H*) under strong directional selection for yellow fluorescence ($\lambda_{ex} = 488$ nm and $\lambda_{em} = 530 \pm 15$ nm, see 'Methods'), allowing only the top 0.5% of cells to survive. We extracted plasmids from the surviving cells in each generation, and used the GFP inserts of these plasmids as templates for the next mutation-selection cycle and for SMRT sequencing. **b**, **c** *H* populations evolved new excitation and emission peaks most slowly and *L* populations did so most rapidly. **b** Relative change in peak excitation during evolution quantified by measuring the fluorescence intensity of evolving populations excited at their optimal excitation peak relative to excitation at 402 nm (note: the optimal excitation peak is 508 nm

for both *L* and *M* populations and 512 nm for *H* populations, whereas it is 402 nm for ancestral GFP; see also Fig. S2a); (**c**) emission peaks of evolving populations in each generation (horizontal axes) (see also Fig. S2b). **d** Specific yellow fluorescence of evolving populations in each generation. The vertical axis indicates specific yellow fluorescence (arbitrary units) for evolving populations in each generation (horizontal axis). We calculated specific yellow fluorescence for each population and time point, dividing yellow fluorescence intensity by the amount of soluble fluorescent protein, as quantified by an ELISA (enzyme-linked immunosorbent assay; see 'Methods'). We performed One-way ANOVAs with Dunnett's post hoc test to ask whether a significant difference existed between *L* and *M*, as well as between *M* and *H* populations in panels **b–d**. Error bars represent one standard error of the mean (SEM) from four replicate populations (colored symbols, see color legend) in panels **b–d**. *$P < 0.05$; **$P < 0.01$; ***$P < 0.001$.

medium or unchanged temperature), and 44 °C (*H* for high temperature). Specifically, we evolved four replicate *E. coli* populations at each temperature, and allowed only the top 0.5% of yellow fluorescing cells in each generation to survive (see 'Methods'). Because we selected directly for yellow fluorescence, we also refer to yellow fluorescence as 'fitness'. In each generation, we generated ~ $10^6$ GFP variants by applying mutagenic PCR to randomly introduce 1.07-1.75 amino acid changing mutations on average into the coding region of GFP (Table S1–S3).

Selection of the new yellow fluorescence phenotype involves a change in both the excitation and emission spectra ($\lambda_{ex}$ = ~402 nm and $\lambda_{em}$ = 512 nm for wild-type GFP vs. $\lambda_{ex}$ = ~510 nm and $\lambda_{em}$ = ~526 nm for evolved YFP). Temperature affected the evolution of both aspects of yellow fluorescence in ways that contradict our initial hypothesis. Specifically, in populations evolved at high temperature the excitation peak shifted most slowly, whereas in populations evolved at low temperature it did so most rapidly (Figs. 1b and S2a). The same holds for the emission peak. In low-temperature populations this peak shifted towards yellow fluorescence one generation earlier than in medium-temperature populations, and two generations earlier than in high-temperature populations (Figs. 1c and S2b).

An improvement in fitness (total yellow fluorescence per cell) can be achieved by either increasing the specific yellow fluorescence (yellow fluorescence intensity per protein molecule) or the number of fluorescing protein molecules in a cell. We consider only the first increase to qualify as phenotypic evolution, because it represents intrinsic single-molecule properties. Our study system allows us to accurately quantify such phenotypic changes during evolution. Specifically, we normalized the phenotypic evolution rate of each evolving population, dividing its yellow fluorescence (fitness) by the amount of soluble fluorescent protein in a cell ('Methods'). Because misfolded proteins are often insoluble, protein solubility can also serve as a proxy for the foldability and stability of proteins at physiological temperatures. We performed this normalization because it excludes the effects of temperature on transcription, translation, and protein folding. Again, *H* populations achieved significantly lower specific yellow fluorescence (yellow fluorescence intensity per protein molecule) than *M* populations during the first four generations of evolution (One-way ANOVA with Dunnett's post hoc test, *P* < 0.05; Fig. 1d). In contrast, *L* populations showed significantly higher specific yellow fluorescence than *M* populations in the first three generations (One-way ANOVA with Dunnett's post hoc test, *P* < 0.01; Fig. 1d). Specific yellow fluorescence leveled off in *L* populations after the third generation, but only reached its maximum in *M* and *H* populations at the last generation (Fig. 1d). In sum, high temperature delays and low temperature accelerates the evolution of yellow fluorescence.

### Elevated temperature slows and lowered temperature accelerates selective sweeps of neofunctionalizing mutations

To understand the genetic causes of phenotypic evolution at different temperatures, we used SMRT sequencing to genotype an average of 2391 fluorescent proteins for each replicate population in every generation (Table S3). We found that thirteen mutations achieved a frequency higher than 20% in at least one replicate *L*, *M*, or *H* population during evolution, and seven of these mutations (F65L, S66G, V164A, I168T, I168V, I172V, and C204Y; Fig. S3) did so in multiple replicate populations (Fig. S4a). Four of the seven high-frequency mutations (S66G, I168T, I168V, and C204Y) achieved a higher frequency in *M* populations than in *H* populations after one round of evolution and three of them reached significantly higher frequency (One-way ANOVA with Dunnett's post hoc test, *P* < 0.001; Figs. 2a and S4b). In addition, the frequency for one of these four mutations remained significantly higher at the end of evolution in *M* populations (I168T; One-way ANOVA with Dunnett's post hoc test, *P* < 0.01; Fig. 2a). In contrast, the frequencies for two of these four mutations (C204Y and I168V) were

significantly lower in *M* populations than in *L* populations (Figs. 2a and S4b). One mutation in particular (C204Y) stood out, because it achieved a significantly higher frequency in *L* populations than in *M* and *H* populations in each generation (One-way ANOVA with Dunnett's post hoc test, *P* < 0.01; Fig. 2a). Moreover, its frequency approached 100% at the end of evolution in all replicate populations, and was much higher than the frequency of any other mutation (Figs. 2a and S4b).

We hypothesized that these four mutations are responsible for the shift to the new phenotype, i.e., they are *neofunctionalizing* mutations[35]. To test this hypothesis, we engineered each of these four mutations into ancestral GFP and determined their excitation and emission spectra. Indeed, all four mutations shifted the excitation peak towards that of yellow fluorescence protein, and two of them (S66G and C204Y) also shifted the emission peak towards yellow fluorescence (Figs. 2b, c, and S5). These data are also supported by previous observations, in which the same four mutations changed excitation and/or emission spectra in the genetic background s of avGFP (mutations 65G, 167T, 167V, and 203Y in avGFP[36–38]; see Fig. S1 for the differences in the coordinate system between our GFP and avGFP). In further support, we found that these four mutations also greatly improved specific yellow fluorescence of our ancestral GFP at all three temperatures (Fig. 2d). Specifically, each of them caused a 3.0–12.1-fold increase in specific yellow fluorescence. C204Y caused an even higher increase than the other three mutations. These observations indicate that the slower evolution of a yellow phenotype in *H* populations resulted from the slower sweep of neofunctionalizing mutations, and especially of C204Y, in *H* populations. Conversely, faster evolution in *L* populations resulted from the faster sweep of these mutations.

Neofunctionalizing mutations often destabilize proteins[39–41]. Our four mutations are no exception: All four neofunctionalizing mutations reduce the melting temperature of GFP significantly (One-way ANOVA with Dunnett's post hoc test, *P* < 0.001; Figs. 2e and S6). This observation raises an important question. How can these mutations be beneficial while reducing protein stability? To find out, we next examined another class of mutations that often occur in experimental and natural evolution. These are mutations that increase the thermodynamic stability or foldability of proteins[3,40,42].

### Stabilizing mutations are most beneficial at high temperature

Among the seven mutations that we had identified as rising to high frequency, three (F65L, V164A, and I172V) did not change the emission and excitation spectra of fluorescent protein, nor did they improve specific yellow fluorescence (Fig. 2b–d). Previous studies have revealed that these three mutations can improve the folding stability of YFP[42] and of avGFP (mutations 64L, 163A and 171V in the coordinate system of avGFP, see also Fig. S1)[38,43]. Absent other phenotypes, we hypothesized that these mutations are beneficial, because they can also increase protein folding stability in the genetic background of our ancestral GFP. To test this hypothesis, we engineered each of the three mutations into ancestral GFP, and measured the melting temperature $T_m$ of the engineered fluorescent protein. Indeed, all three mutations significantly increased this melting temperature (One-way ANOVA with Dunnett's post hoc test, *P* < 0.001; Figs. 2e and S6).

We next examined the evolutionary dynamics of these mutations and found that they reached the highest frequencies in *H* populations, intermediate frequencies in *M* populations, and the lowest frequencies in *L* populations (Figs. 2a and S4b). For example, at the end of evolution, the frequency of variant F65L was 2.4-fold higher in *H* populations than in *M* populations, and 24.2-fold higher in *M* populations than in *L* populations. The second mutation V164A also achieved the highest frequency in *H* populations, lower frequency in *M* populations, and the lowest frequency in *L* populations in each generation of evolution (Fig. 2a). These observations suggest that the stabilizing mutations are

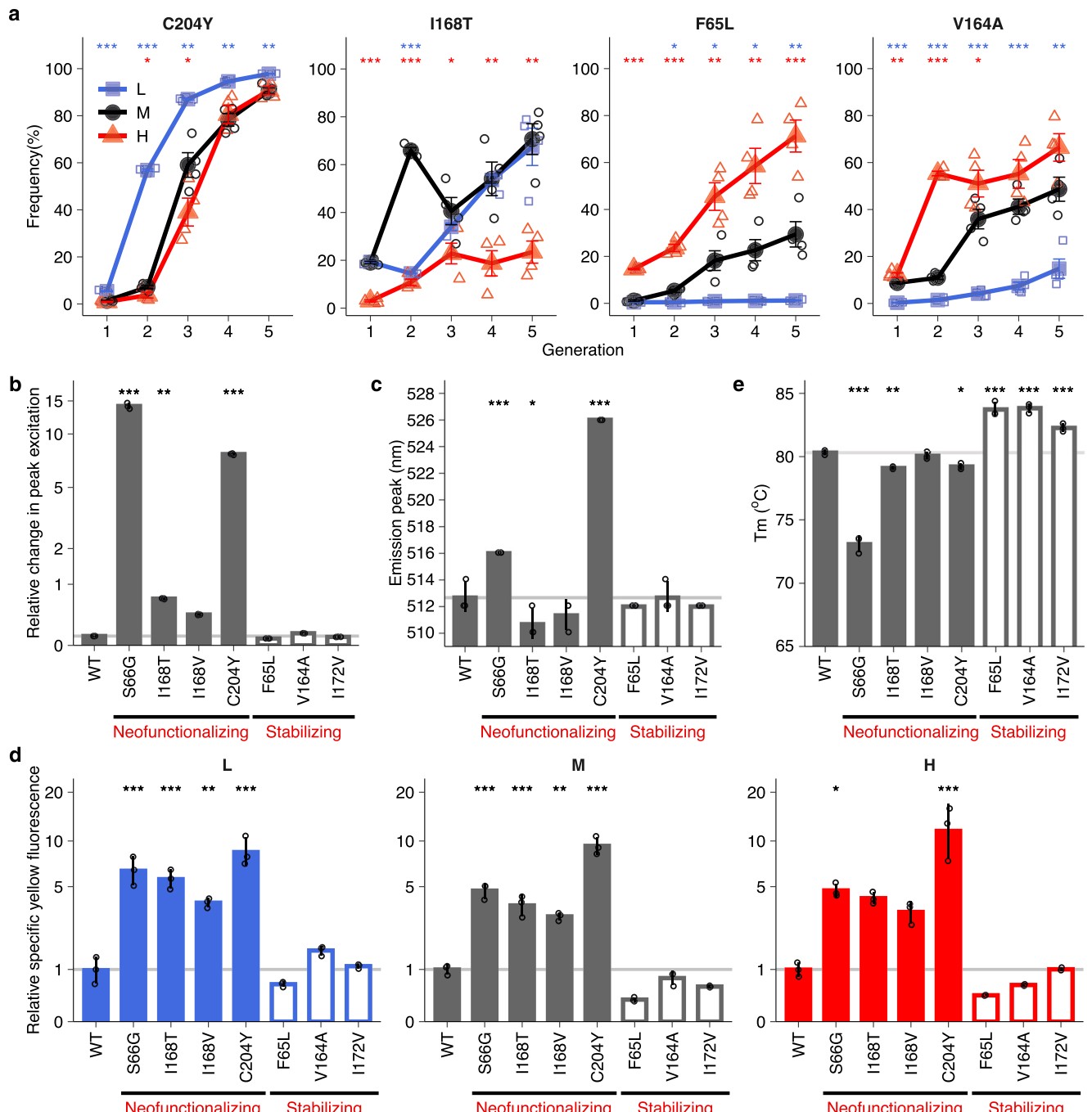

**Fig. 2 | Elevated temperature slows and lowered temperature accelerates selective sweeps of neofunctionalizing mutations. a** Evolutionary dynamics of selected high-frequency mutations in evolving populations during evolution. **b** Four high-frequency mutations (dubbed neofunctionalizing) lead to a change in peak excitation. To estimate the magnitude of the shift in the excitation spectrum, we determined the ratio of fluorescence intensity excited at the mutants' maximum excitation wavelength relative to fluorescence when excited at the ancestral maximum excitation wavelength (vertical axis). **c** Two neofunctionalizing mutations shift the emission peak towards greener fluorescence (see also Fig. S5). **d** Relative specific yellow fluorescence of ancestral GFP (WT) and mutants at low (*L*), medium (*M*) and high (*H*) temperatures. We calculated specific yellow fluorescence by dividing yellow fluorescence intensity by the amount of soluble fluorescent proteins quantified by an ELISA, and did so for ancestral GFP and for each mutant. We then normalized the specific fluorescence for each mutant by dividing it by that of ancestral GFP at the corresponding temperature (see 'Methods'). **e** Melting

temperatures of ancestral GFP (WT) and high-frequency mutants. We estimated the $T_m$ for each mutant by heating cells that expressed the mutant proteins at varying temperatures for 5 min, and then measuring the residual fluorescence (see 'Methods'). This procedure can estimate $T_m$, because the loss of fluorescence after denaturation is mainly caused by exposure of the buried chromophore to an aqueous environment[65]. We performed One-way ANOVAs with Dunnett's post-hoc test to ask whether a significant difference existed between *L* and *M*, as well as between *M* and *H* populations in panel **a**, and to ask whether each mutant was significantly different from ancestral GFP (WT) in panels b-e. Colored symbols (see color legend) and error bars represent one standard error of the mean (SEM) from four replicate populations (colored symbols, see color legend) in panel a. Bar height and error bars represent mean ± SD (one standard deviation) based on three biological replicate fluorescence measurements (open circles) in panels **b**–**e**. *$P < 0.05$; **$P < 0.01$; ***$P < 0.001$.

most beneficial at high temperature and least beneficial at low temperature.

One possible explanation is that these stabilizing mutations increase expression yields by enhancing the transcription and stability of mRNA, or by reducing protein misfolding and insolubilization through slowing translation[33,44]. To test this hypothesis, we engineered all possible synonymous mutations into the codon encoding each of the three stabilizing mutations. The resulting data shows that none of these synonymous mutations substantially changes green fluorescence at either low, medium or high temperature for any of these variants (Fig. S7). Thus, the greater benefit of these mutations at higher temperatures does probably not just result from their effects on transcription and translation.

Another possible explanation is that proteins expressed at a lower temperature are more stable than they need to be – they have excess folding stability – because the slower and less disruptive molecular motions at that temperature are less likely to disrupt a protein fold[45]. This hypothesis creates several predictions. First, our stabilizing mutations should preferentially increase protein solubility at high temperature, but less so at medium or low temperature. This is indeed the case (Fig. 3a). They increase solubility on average 66.4-fold at high temperature, 3.0-fold at medium temperature, but do not significantly affect solubility at low temperature (Fig. 3a). We observed a similar solubility increase in a cell-free expression system that only contained the components necessary for in vitro transcription and translation. Specifically, in this cell-free system all three stabilizing mutations increase solubility on average 10.3-fold at high temperature, but only 2.4-fold at medium temperature and 1.9-fold at low temperature (Fig. S8). The difference in magnitude between the in vivo and in vitro solubility improvement might be due to the fact that the in vivo environment contains chaperones and proteases, which can affect the folding and degradation of different variants to a different extent[40]. Even though the proteins we study are thermally highly stable ($T_m > 70\,°C$), a stabilizing mutation that increases the melting temperature by a few degrees can increase the amount of soluble fluorescent proteins at 44 °C more than 60-fold (Fig. 3a). The reason is that elevated temperature does not just cause inactivation of the mature protein (quantified by $T_m$) but it can also interfere with the post-translational maturation of GFP[46].

Another prediction of the excess-stability hypothesis is that the three stabilizing mutations should be most beneficial at high temperature and least beneficial at low temperature. We tested this by measuring yellow fluorescence at different temperatures. All three stabilizing mutations greatly improved total yellow fluorescence at high temperature, but only one of them (V164A) slightly did so at low temperature (Fig. 3b; note that stabilizing mutations can improve total yellow fluorescence by increasing protein copy number, because ancestral GFP has weak yellow fluorescence). For example, the mutation F65L, which increases the melting temperature by 5.8 °C (Fig. S6b), caused a ~24-fold increase in yellow fluorescence at high temperature (One-sided $t$-test, $P = 6.92 \times 10^{-6}$), but led to a 30.9% decrease of yellow fluorescence at low temperature (One-sided $t$-test, $P = 1.15 \times 10^{-3}$; Fig. 3b). Similarly, both of the stabilizing mutations V164A and I172V greatly increased yellow fluorescence at high temperature (by 73.8-fold and 11.7-fold, respectively) but neither of them did so at low temperature. All of the stabilizing mutations only moderately increased yellow fluorescence at medium temperature (Fig. 3b). In sum, high temperature enhances and low temperature weakens the fitness benefits of stabilizing mutations (Fig. 3c).

### Neofunctionalizing mutations destabilize proteins the most at high temperature but the least at low temperature

The excess stability hypothesis also predicts that fluorescent proteins are most affected at high temperature but least affected at low temperature by the destabilizing effects of neofunctionalizing mutations[45].

We tested this prediction by examining how neofunctionalizing mutations affect protein solubility. Two of the four mutations reduced protein solubility at high temperature (Fig. 3a), a reduction that was greatest (75.5%) for the neofunctionalizing mutation C204Y (One-sided $t$-test, $P = 0.03$; Fig. 3a). In contrast, only one of the four mutations reduced protein solubility (and to a lesser extent) at medium temperature (60.6%). None of the four mutations reduced solubility significantly at low temperature (Fig. 3a).

A further prediction is that neofunctionalizing mutations preferentially increase fitness at low temperature. Indeed, while all four neofunctionalizing mutations enhanced yellow fluorescence greatly, they did so to different extents across the three temperatures (Fig. 3b). Three of them increased yellow fluorescence to a greater extent at low temperature than at high temperature (Fig. 3c). Mutation C204Y led to the greatest increase in yellow fluorescence at low temperature (13.4-fold). Importantly, it improved yellow fluorescence to a greater extent at low temperature than at medium and high temperatures (One-way ANOVA with Dunnett's post hoc test, $P < 0.001$; Figs. 3b and S9a). Another neofunctionalizing mutation, I168T, which achieved a 2.9-fold higher frequency in $L$ populations than in $H$ populations at the end of evolution (Fig. 2a), was also more beneficial at low temperature than at medium and high temperatures (Figs. 3b and S9a). These observations confirm that elevated temperature weakens and lowered temperature enhances the fitness benefits of neofunctionalizing mutations.

The excess stability hypothesis also suggests that stabilizing mutations should outcompete neofunctionalizing mutations during evolution in $H$ population, and that the opposite should be the case in $L$ populations. Indeed, we found that the mean number of stabilizing mutations per GFP coding sequence was 1.1–2.4-fold higher than that for neofunctionalizing mutations in $H$ populations during the first three generations, but it was 9.8–41.9-fold lower in $L$ populations during each generation of evolution (Fig. 3d). Taken together, these observations indicate that elevated temperature exacerbates the destabilizing effects and reduces the fitness benefits of neofunctionalizing mutations, whereas lowered temperature reduces their destabilizing effects and thus magnifies their benefits.

### Epistatic interactions promote the spreading of stabilizing mutations at high temperature

Neofunctionalizing and stabilizing mutations may interact non-additively when they occur on the same molecule. Such non-additive (epistatic) interactions can be positive or negative, depending on whether two adaptive mutations together lead to a fitter or less fit genotype than expected. To understand how epistasis may affect phenotypic evolution, we first identified genotypes with multiple amino acid mutations in our populations. Only three double mutants (and no higher order mutants) achieved a frequency exceeding 50% in $L$, $M$ or $H$ populations at the end of evolution (C204Y + I168T, C204Y + F65L, and C204Y + V164A; Fig. 4a).

The first double mutant (C204Y + I168T) consists of two destabilizing neofunctionalizing mutations (Fig. 2d, e). It caused a much smaller increase in yellow fluorescence at high temperature (3.9-fold increase) than at medium and low temperatures (8.1-fold and 35.2-fold increase, respectively; Fig. 4b). Though part of this difference is caused by a lower improvement in yellow fluorescence at high temperature for each of its constituent mutations (Fig. S9a), we wondered whether high temperature can also alter interactions between these mutations and thus further contributes to the double mutant's lower fitness increase. To find out, we calculated the extent of epistasis between mutation pairs based on the expression $e = (F_{A+B} - F_{WT}) - (F_A - F_{WT}) \cdot (F_B - F_{WT})$, where $F_{A+B}$, $F_A$, $F_B$, and $F_{WT}$, respectively, represent the logarithmically transformed yellow fluorescence of the double mutant A + B, of its constituent mutants A and B, and of ancestral GFP[47]. The extent of epistasis $e$ represents the relative difference between the fitness effect of the double mutant and the sum of the effects of both constituent

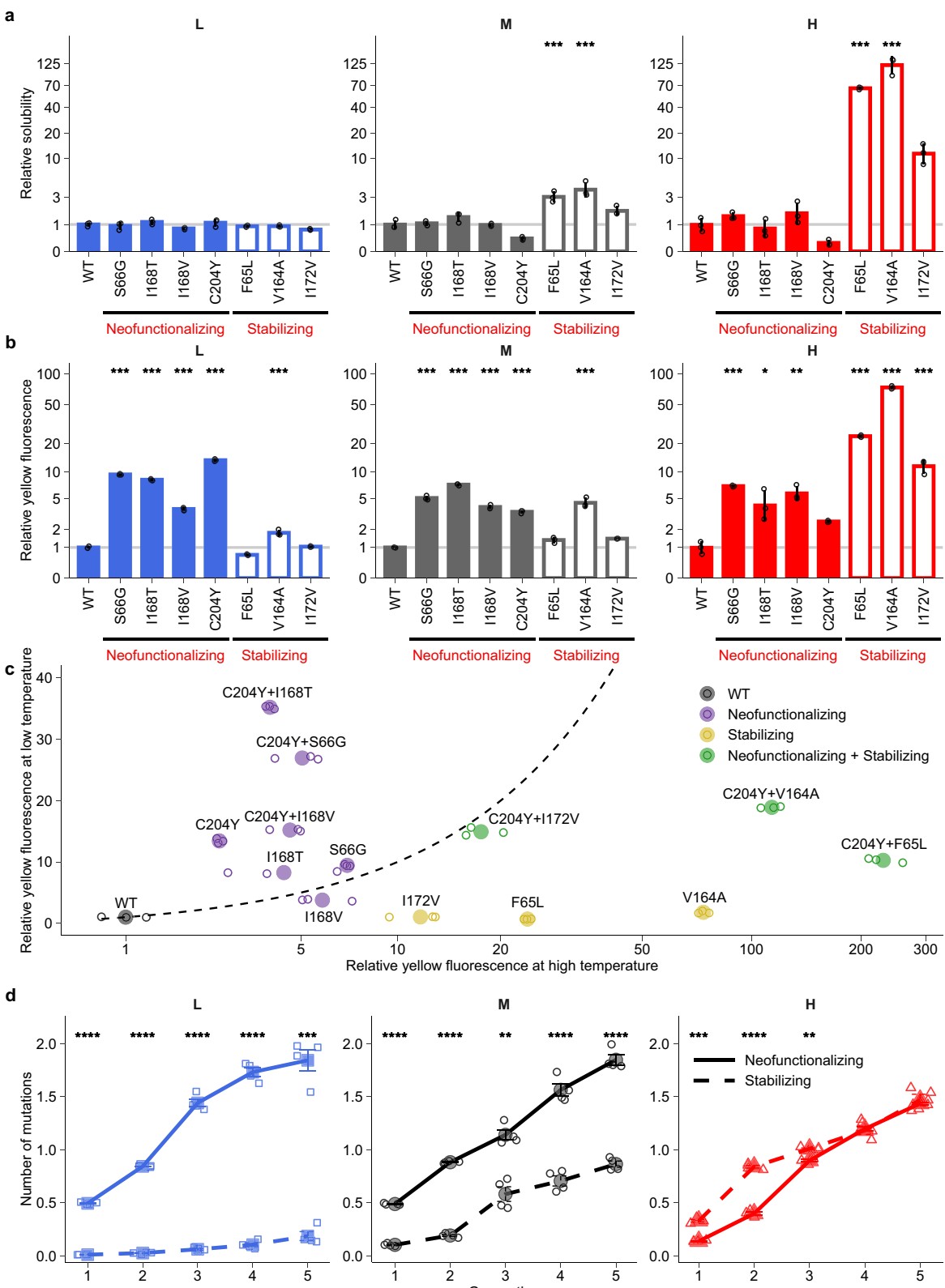

mutations. A negative (positive) value of $e$ indicates negative (positive) epistasis, i.e., the combined effects of the mutations are smaller than (greater than) expected from additivity. A value of $e = 0$ indicates no epistasis, i.e., additivity. At all temperatures, the extent of epistasis was far below zero, indicating that the double mutant C204Y + I168T increases fitness less than the sum of its constituent single mutants (Fig. 4c). However, at high temperature epistasis was less negative than

at medium and low temperatures (Fig. 4c). Consistent with this observation, I168T did not significantly reduce the solubility of C204Y at all temperatures (Fig. 4d). This demonstrates that the lower fitness improvement for the double mutant C204Y + I168T at high temperature mainly resulted from a smaller increase in yellow fluorescence for each of its constituent mutations rather than from negative epistasis between them. The resulting lower fitness improvement caused a much

**Fig. 3 | Elevated temperature reduces and lowered temperature enhances the beneficial effects of neofunctionalizing mutations. a, b** Relative solubility (**a**) and relative yellow fluorescence (**b**) of single mutants at low (*L*), medium (*M*) and high (*H*) temperatures. The vertical axes indicate the amount of soluble protein (**a**) or the fluorescence intensity (**b**) for each mutant (horizontal axes) relative to ancestral GFP. **c** Neofunctionalizing mutations and stabilizing mutations are more beneficial at low and high temperatures, respectively. The horizontal (*x*) and vertical (*y*) axes indicate yellow fluorescence of single and double mutants relative to ancestral GFP at low (*L*) and high (*H*) temperatures based on three biological replicate measurements (single small symbols). The dashed line indicates equality (*y* = *x*), and values above (below) this line indicate the mutants are more (less) beneficial at low temperature. Colors indicate ancestral GFP (WT, gray), mutants that carry neofunctionalizing mutations (purple), stabilizing mutations (yellow), or both neofunctionalizing and stabilizing mutations (green). **d** Evolutionary

dynamics of neofunctionalizing mutations and stabilizing mutations. The vertical axes indicate the mean number of neofunctionalizing (solid line) or stabilizing (dashed line) mutations per sequence in each generation of evolution (horizontal axes). Bar height and error bars represent mean ± SD based on three biological replicate measurements (single small symbols) in panels **a** and **b**, and mean ± SEM based on four replicate populations (circles, squares, or triangles) in panel **d**. Note the logarithmic vertical scales in panels **a** and **b**. We performed One-way ANOVAs with Dunnett's post hoc test to ask whether fluorescence or solubility of mutants was significantly different from that of ancestral GFP (WT) in panels **a** and **b**. We used two-sided *t*-tests with 'Holm' adjustment to test whether the average number for neofunctionalizing mutations per sequence was significantly different from that for stabilizing mutations in *L, M,* and *H* populations in panel **d**. *$P < 0.05$; **$P < 0.01$; ***$P < 0.001$; ****$P < 0.0001$.

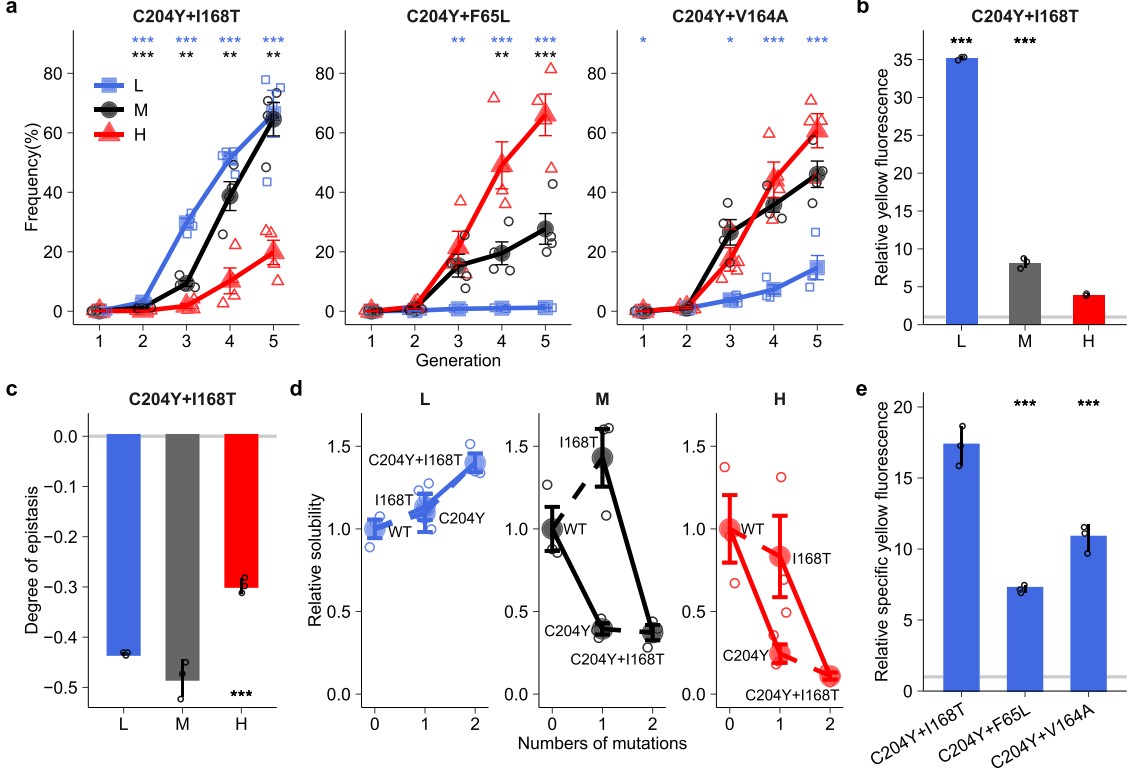

**Fig. 4 | A high-fitness genotype, harboring only neofunctionalizing mutations, sweeps most rapidly at low temperature. a** Frequency dynamics of high-fitness double mutants during evolution at low (L), medium (M) and high (H) temperatures. **b** Relative yellow fluorescence of C204Y + I168T at different temperatures. **c** Negative epistasis between the constituent mutations of C204Y + I168T at different temperatures. **d** Relative solubility of C204Y + I168T and its constituent mutations at different temperatures. **e** Relative specific yellow fluorescence of three high-fitness double mutants at low temperature. The vertical axes in panels **b** and **d**, respectively, indicate the yellow fluorescence intensity and solubility of each mutant (horizontal axes) relative to ancestral GFP at the corresponding temperature. The vertical axis in panel **c** indicates the degree of epistasis (see text for details) between the constituent mutations of C204Y + I168T. The vertical axis in

panel **e** indicates the specific yellow fluorescence of each double mutant (horizontal axis) relative to ancestral GFP at low temperature. We calculated specific yellow fluorescence by dividing yellow fluorescence intensity by the amount of soluble fluorescent proteins, and then normalized the specific yellow fluorescence for each mutant by dividing it by that of ancestral GFP at low temperature (see 'Methods'). Bar height and error bars represent mean ± SEM based on four replicate populations for panel a, and mean ± SD based on three biological replicate measurements (single small symbols) for panels **b–e**. We performed One-way ANOVAs with Dunnett's post hoc test in panels **a-c** and **e**. In panel **d**, solid lines between ancestral GFP (WT) and a mutant or between mutants indicate a significant difference in relative solubility between them ($P < 0.05$, One-sided *t*-test) and dashed lines indicate no significant difference. *$P < 0.05$; **$P < 0.01$; ***$P < 0.001$.

slower sweep of the genotype C204Y + I168T through all *H* populations than through *M* or *L* populations (Fig. 4a). This slower spreading contributed to the slower phenotypic evolution in *H* populations, because the genotype conveyed much higher specific yellow fluorescence (>1.6-fold) than the other two double mutants C204Y + F65L and C204Y + V164A that swept most rapidly through *H* populations (One-way ANOVA with Dunnett's post hoc test, $P < 0.001$; Fig. 4a, e).

The remaining double mutants (C204Y + F65L and C204Y + V164A) combine the neofunctionalizing (but destabilizing) mutation

C204Y with one of the two stabilizing mutations F65L and V164A (Fig. 2e). For the double mutant C204Y + F65L, high and medium temperatures caused a positive interaction between its single constituent mutations, resulting in a higher fitness effect than expected from the fitness effects of two single mutants (Fig. 5a). This positive epistasis was significantly greater at high temperature than at medium temperature (One-way ANOVA with Dunnett's post hoc test, $P < 0.001$; Fig. 5a). In contrast, low temperature almost completely suppressed this positive epistasis and caused the interaction to become additive

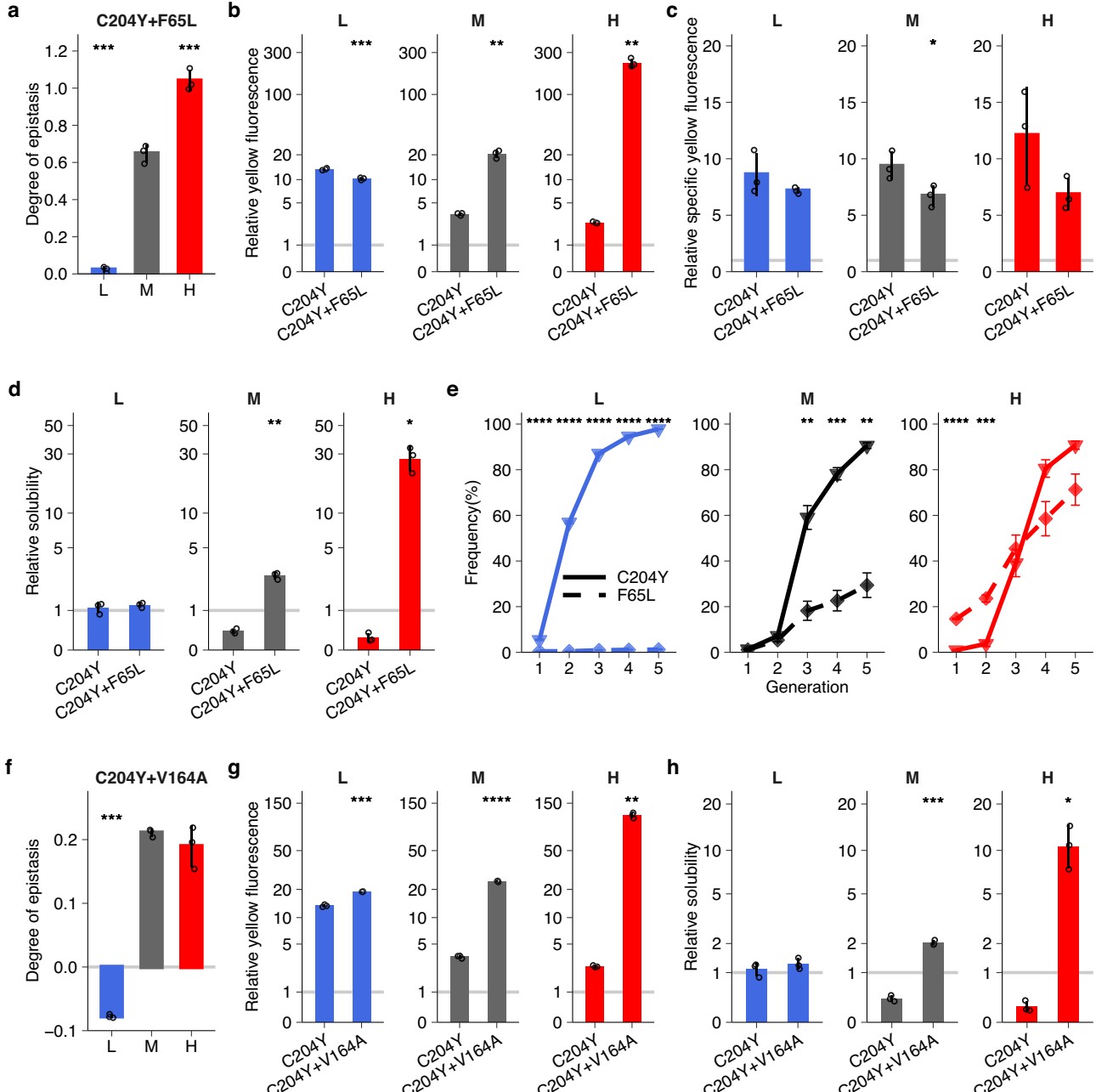

**Fig. 5 | Epistasis accelerate sweeps of stabilizing mutations at high temperature. a** High temperature enhances and low temperature neutralizes positive epistasis between the constituent mutations of C204Y + F65L. **b–d** Effects of F65L on yellow fluorescence (**b**), specific yellow fluorescence (**c**) and solubility (**d**) at the genetic background of C204Y at low (L), medium (M) and high (H) temperatures. **e** Frequency dynamics of C204Y and F65L during evolution at different temperatures. **f** Increasing temperature changes epistasis between the constituent mutations of C204Y + V164A from negative to positive. **g, h** Effects of V164A on yellow fluorescence (**g**) and solubility (**h**) at the genetic background of C204Y at different temperatures. The vertical axes in panels **a** and **f** indicate the degree of epistasis (see text for details) between the constituent mutations of C204Y + F65L and of C204Y + V164A, respectively. The vertical axes in panels **b–d** indicate the yellow fluorescence intensity, specific yellow fluorescence, and solubility of C204Y and

C204Y + F65L relative to ancestral GFP at the corresponding temperature. The vertical axis of panel e indicates the frequency of each mutant at the corresponding temperature. We calculated specific yellow fluorescence by dividing yellow fluorescence intensity by the amount of soluble fluorescent proteins and then normalized the specific yellow fluorescence for each mutant by dividing it by that of ancestral GFP at the corresponding temperature (see 'Methods'). The vertical axes in panels g and h indicate the yellow fluorescence intensity and solubility of C204Y and C204Y + V164A relative to ancestral GFP at the corresponding temperature. Bar height and error bars represent mean ± SD based on three biological replicate measurements (single small symbols) for panels **a–d** and **f–h**. We performed one-way ANOVAs with Dunnett's post hoc test in panels **a** and **f**, two-sided *t*-tests in panels **b–d** and **g, h**, and two-sided *t*-tests with 'Holm' adjustment in panel e. *$P < 0.05$; **$P < 0.01$; ***$P < 0.001$; ****$P < 0.0001$.

(Fig. 5a). In addition, the stabilizing mutation F65L greatly increased yellow fluorescence of C204Y at high temperature (88.4-fold) and medium temperature (5.9-fold) but significantly reduced it at low temperature (Two-sided *t*-test, $P < 0.001$; Fig. 5b). This switch from

being beneficial at high and medium temperatures to deleterious at low temperature originates from the extent to which F65L can compensate for the destabilizing effect of C204Y. Because F65L reduces specific yellow fluorescence by 16.7–43.4 % at all temperatures when

introduced into the C204Y mutant (Fig. 5c), it can achieve a net fitness benefit only by increasing solubility. This is the case at high temperature, where F65L increases solubility by 116.4-fold (Fig. 5d). At medium temperature, it still increases solubility by 6.9-fold. In contrast, at low temperature, it no longer significantly increases solubility (Fig. 5d). In consequence, increased solubility cannot compensate for F65L's reduction in specific fluorescence at low temperature. These temperature-specific effects are also underscored by the evolutionary dynamics of the constituent mutations. At high temperature, F65L rose to high frequency before C204Y did (Fig. 5e), which is consistent with its stronger fitness benefit at this temperature (Fig. 3a). At medium temperature, F65L is less beneficial than C204Y, and it thus also rose to a high frequency more slowly than C204Y. Finally, at low temperature, the frequency of F65L stayed below 1.2% during evolution (Fig. 5e). Similarly, high temperature led to greater positive epistasis between the constituent mutations of the double mutant C204Y + V164A than medium temperature but low temperature resulted in slightly negative epistasis (Fig. 5f), consistent with the observations that the stabilizing mutation V164A causes a higher increase in yellow fluorescence on top of C204Y at high temperature than at medium and low temperatures (Fig. 5g), and that it can greatly compensate for the destabilizing effect of C204Y at high temperature (Fig. 5h). By enhancing positive epistasis between C204Y and F65L or V164A, high temperature made both double mutants more beneficial than C204Y + I168T (Fig. S10) which only harbors neofunctionalizing mutations and has higher specific yellow fluorescence (Fig. 4e). As a result, high temperature helped these two genotypes outcompete C204Y + I168T (Fig. 4a), and thus delayed the evolution of the new yellow fluorescence phenotype (Fig. 1).

In sum, high temperature slows phenotypic evolution by favoring stabilizing mutations and enhancing their positive epistasis with neofunctionalizing mutations. In contrast, low temperature accelerates phenotypic evolution by favoring destabilizing neofunctionalizing mutations, and by reducing or eliminating their positive epistasis with stabilizing mutations.

## Discussion

Our observations demonstrate that increasing the temperature at which a protein evolves can hinder phenotypic adaptation. In contrast, reducing temperature can accelerate such adaptation (Fig.1). Increasing the temperature worsens the destabilizing effects of neofunctionalizing mutations and thus slows their sweeps (Figs. 2a and 3c). Conversely, lowering the temperature creates excess protein folding stability (Figs. 2e and 3a), which reduces the destabilizing cost of neofunctionalizing mutations, increases their fitness benefit, and thus helps them spread faster through a population (Figs. 2a, 3c, d, and 4a).

Protein stability is crucial for protein evolution[31,48,49] and is a major goal in protein engineering[50,51]. Increasing temperature destabilizes proteins by accelerating molecular motions, whereas decreasing temperature stabilizes proteins by slowing such motions[45]. Everything else being equal, temperature thus changes the equilibrium between folded and unfolded protein states. In other words, a temperature increase can reduce the amount of correctly folded and thus soluble fluorescent proteins. For example, we found that increasing the temperature from 37 °C to 44 °C led to a 17.2-fold decrease of fluorescent protein solubility – a measurable proxy for protein stability (Fig. S9b). This stability loss can be rescued by stabilizing mutations (Fig. 3a). Such mutations become especially beneficial during evolution at high temperature, where they benefit fitness more than neofunctionalizing mutations (Fig. 3b). In consequence, they spread more rapidly than neofunctionalizing mutations at high temperature (Fig. 3d), which delays phenotypic adaptation (Fig. 1).

At any one location on Earth, temperature may rapidly fluctuate by many degrees due to diurnal cycles and seasonal changes. Such temperature changes can alter the strength of selection to modify

protein stability during the evolution of new protein functions. Our experiments show that an increase in temperature strengthens and a decrease weakens selection for stability (Fig. 3d). Because adaptive mutations that bring forth new protein functions usually destabilize proteins[39–41,52], stronger selection on protein stability at high temperatures can delay the spreading of neofunctionalizing mutations through an evolving population. Consequently, increasing temperature can slow down and lowering temperature can speed up the evolution of a new phenotype (Fig. 1). We expect that our observations will apply whenever neofunctionalizing mutations reduce the stability of adaptively evolving proteins. Whether they also apply when neofunctionalizing mutations do not reduce stability[35] is an exciting question for future work.

Populations of organisms cope with elevated temperatures through evolutionary adaptation both in the laboratory and in the wild[13–15,53–56]. However, once they have exceeded their upper thermal limits they often go extinct[15,54,57]. Our observations may help explain such extinction. First, evolutionary rescue becomes impossible if high temperature destabilizes important proteins so much that stability-enhancing mutations become exceedingly rare. Second, if the emergence or improvement of an adaptive phenotype necessary for survival requires one or more strongly destabilizing mutations, a temperature increase may render the mutant proteins unable to fold, and thus prevent the rescue of the population through adaptive evolution[15,22].

Our work has several practical implications. For example, it suggests that lowering temperature substantially below that required for soluble protein expression can help evolve or engineer proteins with new functions, because it helps function-altering mutations to manifest their beneficial effects. This strategy may be especially helpful when conventional approaches fail, because neofunctionalizing mutations are too strongly destabilizing[41,58,59]. Another implication relates to ongoing global climate change, which can exert severe thermal stress on organisms and force them to undergo adaptive evolution to higher temperatures[60,61]. Although global climate change itself alters the average global environment only very gradually, one of its consequences is an increase of extreme climatic events, such as extreme heat waves with rapid temperature increase[48,61–63]. Such a rapid temperature increase can purge neofunctionalizing mutations that destabilize proteins. Consequently, it may impede phenotypic evolution, and thus result in decreased genetic diversity and biodiversity. Most generally, our work suggests that a temperature change can profoundly affect how life finds a way towards phenotypic innovation.

## Methods
### Strains and plasmids
We used the plasmid pBAD202/D-TOPO (K4202-01, Invitrogen) as our expression vector. This plasmid harbors a Kanamycin-resistance gene and an arabinose-inducible araBAD promoter that controls the expression of a GFP gene, which is integrated into the vector between *Xho*I and *Hind*III restriction sites, as described previously[34]. We refer to the encoded GFP protein as our 'ancestral' GFP. It carries a methionine insertion at the second residue and six amino acid substitutions compared to the *Aequorea victoria* green-fluorescent protein (avGFP) (Fig. S1). It has an excitation maximum at 405 nm, and an emission peak at 512 nm. We used the *E. coli* host BW27783 (CGSC 12119) throughout to ensure homogeneity of GFP expression.

### Preparation of electrocompetent cells
We inoculated a single colony of *E. coli* strain BW27783 into 5 mL SOB medium in a 50 mL flask, and grew the resulting culture at 37 °C with shaking at 220-250 rpm for 12–16 h. After overnight incubation, we transferred 3 mL of the culture into 300 mL SOB medium in a 2 L flask, and continued the incubation until the $OD_{600}$ value equaled 0.4–0.6

(1 cm optical path length). We then placed the culture on ice for at least 15 min, and pelleted cells at 4 °C by centrifugation at 1500 g for 15 min. We resuspended the cell pellets using 60 mL ice-cold ddH$_2$O and distributed them equally into three 50 mL tubes. We then used a 10 mL pipette to slowly deliver 10 mL of ice-cold glycerol/mannitol solution (20% glycerol (w/v) and 1.5% mannitol (w/v)) to the bottom of each tube. We set the centrifuge's acceleration/deceleration to zero and used it to pellet cells by centrifugation at 1500 g and 4 °C for 15 min by a concentrator (Eppendorf 5810/5810R). We removed the supernatant by aspiration and resuspended cell pellets in 3.0–4.0 mL ice-cold glycerol/mannitol solution. We used a dry ice-ethanol bath to freeze the resulting cell suspensions for ~1 min, and then stored at −80 °C for the following electroporation experiments.

## Preparation of mutant libraries and expression of fluorescent proteins

In every generation, we inserted mutated GFP coding regions into fresh backbones and transformed into fresh competent cells to avoid the incidence of mutations in other region of plasmid or the genome of *E. coli*. We conducted mutagenic PCR to introduce random mutations into the coding region of GFP, as previously described[64], but with the following modifications. In brief, we prepared 50 μL of a PCR reaction mix that consists of 35.25 μL ddH$_2$O, 2.5 μL template (1 ng/μL), 5 μL of 10×ThermoPol buffer (M0267L, NEB), 0.25 μL of *Taq* DNA polymerase (5 U/μL, M0267L, NEB), 2 μL of dNTPs (10 mM, R0192, Thermo Scientific), 1.5 μL of 8-oxo-GTP/dPTP (100 μM, Trilink Biotechnologies), and 1 μL of each primer (Mutafp511F - GAAGGAGATATACCTCGAG/ Mutafp511R - AGACCGTTTAAACAAGCT T).

We executed the following thermocycling program to perform the PCR: 95 °C/30 s; 20 cycles of 95 °C/20 s, 46 °C/30 s and 68 °C/ 6 min. After PCR amplification, we used a QIAquick PCR purification kit (QIAGEN) to purify PCR products, added 1 μL of *Dpn*I (R0176S, NEB), and incubated at 37 °C for ~1 h to remove the template plasmid. Then we added 2 μL *Xho*I and *Hind*III (R0146L/R3104S, NEB), and continued the incubation at 37 °C to digest the purified PCR products. After overnight digestion, we purified the mutated GFP pools as inserts for ligation by using the QIAquick PCR purification kit.

We performed the ligation reaction in a 20 μL solution, which contained ~60 ng of purified inserts, ~100 ng of purified vector backbone, 10 U of *T4* DNA ligase and 1×Ligation buffer (M0202L, NEB). We performed the ligation reaction at 20–22 °C for ~16 h, and subsequently purified the ligation product by precipitation. To this end, we mixed the ligation product with 80 μL of ddH$_2$O, 50 μL of 7.5 M ammonium acetate (A2706-100 mL, Sigma), 1 μL of glycogen (R0551, Thermo Scientific), and 375 μL of ice-cold absolute ethanol. We kept the mixture in a −80 °C freezer for ~20 min, and then pelleted the ligated plasmids by centrifugation at 18,000 g for 20 min. We washed the pellet once using 800 μL of cold ethanol (70%), and dried it using a concentrator (Eppendorf 5301). We then used 10 μL of ddH$_2$O to dissolve the pellet as the purified ligation product for electroporation.

We mixed 5 μL of the resulting ligation product with 100 μL of electrocompetent BW27783 cells, and transferred the mixture into a 0.2 cm pre-cooled cuvette (EP202, Cell Projects, UK). We used a Micropulser electroporator (Bio-Rad) to perform electroporation at 15 kV/cm. Then we immediately added 1 mL of pre-warmed SOC medium and transferred the resulting culture into a 50 mL tube and incubated it for 1.5 h at 37 °C with shaking at 220 rpm. We added 10 mL of LB containing 50 μg/mL kanamycin. We serially diluted the resulting transformants using saline, and plated 50 or 100 μL of the serially diluted aliquot on LB agar supplemented with 25 μg/mL of kanamycin to estimate the mutant library size, which equaled ~10$^6$ colonies per transformation. We continued to incubate the remaining culture for 12 - 14 h. We then sampled 2 mL of this culture and pelleted cells by centrifugation at 9000 g and 4 °C for 5 min. We resuspended cells with

2 mL LB containing 50 μg/mL kanamycin and 0.2% arabinose in a 14 mL tube. We incubated the resulting culture using a microplate shaker with a shaking speed of 480 rpm (VWR) at 25 or 37 °C for ~12 h, or 44 °C for ~8 h. After induction, we put the culture on ice until we performed cell sorting.

## Sorting cells

We selected high-fluorescence cells by using a cell sorter (BD Aria III) at ~25 °C for all evolving populations, because this instrument does not allow us to change the temperature of droplets harboring cells when quantifying fluorescence. However, the fluorescence detected at ~25 °C for a given variant is likely to reflect its fluorescence relative to ancestral GFP or to other variants at 37 °C for *M* populations and at 44 °C for *H* populations. This is because GFP and its variants are highly thermostable (usually $T_m > 60$ °C) once they have folded correctly and the chromophore has matured. In other words, protein folding and the maturation of the chromophore at different temperatures are the key factors that determine the fluorescence of variants for our *L*, *M*, and *H* populations. This is evident from our observation that the fluorescence intensities for all seven high-frequency mutants relative to ancestral GFP did not change substantially when measured at 26, 36, and 42 °C (Fig. S11). In sum, our study system enables us to minimize the interference of potential factors other than temperature on a protein's phenotype.

After induction of GFP expression at different temperatures, we mixed 20–40 μL of culture with 800 μL of PBS buffer, and used the resulting cell suspension for sorting. Specifically, we selected the top 0.5% of yellow fluorescing cells (Fig. 1a) with an Aria III cell sorter (BD Biosciences) in the FITC channel ($λ_{ex} = 488$ nm, $λ_{em} = 530 ± 15$ nm). We sorted cells at 4 °C and collected 10$^4$ cells in ~1 mL LB medium in a 1.5 mL tube for every population grown at 25 and 37 °C, as well as 2 × 10$^4$ cells for populations grown at 44 °C. We selected two-fold more cells for every population that grew at 44 °C because cell viability for populations grown at 44 °C is roughly half that of populations grown at 25 and 37 °C. We put the selected cells on ice until we had sorted all samples to avoid cell death or proliferation. After sorting all samples, we incubated the selected cells at 37 °C with a shaking speed of 220 rpm for 3 - 5 h. We then mixed the culture with 2 mL LB supplemented with 75 μg/mL kanamycin and continued the incubation for ~12 h. We sampled 1 mL of overnight culture for isolating plasmids by using a QIAprep Spin Miniprep Kit (Qiagen). We used the isolated plasmids as templates for the next-round of PCR mutagenesis, as well as for SMRT sequencing. In addition, we mixed 0.9 mL of overnight culture with 0.6 mL 50% glycerol to make a glycerol stock stored at −80 °C for the further experiments described below.

## Engineering GFP mutants

We engineered single mutants and double mutants by using whole plasmid PCR. Specifically, we designed a pair of primers for each single or double mutant that carried the corresponding mutation. To generate all single mutants, we used ancestral GFP as the template for whole plasmid PCR. To engineer double mutants that all harbored the mutation C204Y, we used the mutant C204Y as the template for a whole plasmid PCR. We performed the whole plasmid PCR in a 50 μL reaction solution which contains 1 ng of template plasmid, 10 μL of 5×Q5 reaction buffer, 10 μL of 5×Q5 high GC enhancer, 1.0 μL of 10 nM dNTPs, 1.0 μL of each primer (10 μM), 0.5 μL of Q5 High-Fidelity DNA Polymerase, and 25.5 μL of ddH$_2$O. We executed the PCR with the following program: 98 °C/ 30 s; 8 cycles of 98 °C/15 s, 64 or 72 °C/10 s and 72 °C/2 min; 20 cycles of 98 °C/15 s and 72 °C/130 s; 72 °C/5 min. We purified the PCR products using the QIAquick PCR purification kit and used *Dpn*I to remove the template plasmid by digesting the PCR product at 37 °C for 1.5 - 2 h. We sampled 1 μL of the digested PCR product, and mixed it with 30 μL electrocompetent cells for electroporation. After allowing the transformants to recover at 37 °C for ~1 h, we plated

50 μL of recovered transformants on LB agar supplemented with 25 μg/mL kanamycin. We picked three to six colonies from each transformation and Sanger-sequenced the plasmid DNA. We chose a correct construct for each mutant, confirmed by Sanger sequencing, and grew it in 2 mL LB (50 μg/mL kanamycin) at 37 °C with shaking at 220 rpm. Then we mixed 600 μL 50% glycerol with 900 μL overnight culture and stored the mixture at −80 °C for the further experiments. We isolated the plasmids and used a *Xho*I and *Hind*III digestion for further confirmation. To add a 6 × His tag at the C-terminal of ancestral GFP and each mutant, we used the correctly constructed plasmids as templates to perform whole plasmid PCR using the following pair of primers (HindHISF- TACAAGAAGCTTCATCATCACCATCACCATtgaG TTTAAACGGTCTCCAGCTTGGCT/ HindHISR- AACTCAATGGTGATGG TGATGATGAAGCTTCTTGTACAGCTCGTCCATGCCGAGAG. The PCR followed the same procedure we have just described. We used the resulting His-tagged ancestral GFP and its mutants for further experiments.

We followed the same procedure as described above to engineer synonymous mutations for the mutants F65L, V164A, and I172V by whole plasmid PCR. We confirmed the correct construction for each mutant by Sanger sequencing and prepared glycerol stocks for further experiments, as described above.

### Fluorescence assay, excitation/emission spectrum scan, and quantification of soluble proteins

To perform a fluorescence assay for each replicate evolving population, we transferred 20 μL of glycerol stock into 2 mL LB medium supplemented with 50 μg/mL kanamycin, and grew the resulting culture at 37 °C in a VWR microplate shaker (VWR International, USA) with a shaking speed of 480 rpm or in a WIGGENS microplate shaker with a shaking speed of 480 rpm (WIGGENS GmbH, Germany) for ~12 h. We sampled 1 mL of culture and pelleted cells by centrifugation at 9000 g and 4 °C for 5 min. We resuspended cells in 2 mL LB medium containing 50 μg/mL of kanamycin as well as 0.2% arabinose, and continued the incubation for ~12 h at 25 and 37 °C, as well as for ~10 h at 44 °C. We then transferred 20 μL of the culture into 190 μL PBS, and mixed thoroughly in a 96-well plate. Subsequently, we added 10 μL of the resulting cell suspension to another 190 μL PBS for quantifying fluorescence intensity at the single-cell level (Fig. S12), using a Fortessa (BD Biosciences) or CytoFLEX LX cell analyzer (Beckman Coulter). We used the remainder of the cell suspension for measuring fluorescence intensity, and scanned the excitation/emission spectrum using a Tecan plate reader (Spark). We also sampled 1 mL of culture to extract soluble proteins by using a CelLytic™ B Cell Lysis Reagent (B7435-500 mL, Sigma) or a One-Step Bacterial Active Protein Extraction Kit (Sangon Biotech). We followed the manufacture's protocol to quantify the amount of soluble fluorescent proteins by using a GFP ELISA Kit (AKR-121, Cell Biolabs Inc.), which can detect GFP, BFP, CFP, and YFP from *Aequorea Victoria*. We followed the same procedure to perform fluorescence assays, excitation/emission scans, and quantification of soluble proteins for each of the His-tagged engineered mutants, except that we used a His-tag ELISA Detection Kit (L00436, Genscript) rather than a GFP ELISA Kit to quantify soluble protein.

### Protein thermal stability assay

To estimate the thermal stability of each mutant and ancestral GFP, we thoroughly mixed 1 μL of crude lysate with 99 μL of TNG (100 mM Tris, 100 mM NaCl, 10% glycerol, 10 mM DTT, 1×cOmplete™ (EDTA-free Protease Inhibitor Cocktail, Roche 11873580001), pH 7.5) buffer by pipetting. We then used a thermocycler to heat the resulting mixture for 5 min in a temperature range of 60–80 °C (60.0, 60.6, 62.0, 64.1, 66.6, 69.0, 71.6, 74.0, 76.5, 78.6, and 80.0 °C), followed by a 30-second incubation at 4 °C. Immediately after the incubation, we transferred 90 μL of each mixture to a 96-well microplate, and used a Tecan Spark

plate reader ($\lambda_{ex}$ = 400 nm, $\lambda_{em}$ = 512 nm) to measure its fluorescence intensity. We used the unheated lysate-buffer mixture and the GFP-free lysate (from cells without a GFP gene) as positive and blank controls, respectively. We calculated the residual fluorescence as fluorescence relative to the positive control. We then used the resulting data to fit a sigmoidal model, and derived the midpoint temperature of the transition curve ($T_m$, the temperature at which 50% of proteins unfold) for each mutant, as well as for ancestral GFP (WT) (Fig. S6a). This procedure can estimate the denaturation midpoint of each mutant and of ancestral GFP by directly measuring the green fluorescence change, because the chromophore of fluorescent proteins still stays chemically intact even in the denatured state, and the loss of fluorescence after denaturation is mainly caused by exposing the buried chromophore to an aqueous environment[65].

In addition, we also measured the in vivo $T_m$ for ancestral GFP and its variants. Specifically, to quantify $T_m$ for any one of these proteins, we first grew cells for inducible protein expression at 37 °C, as described above. We then sampled ~300 μL of cell culture and pelleted the cells by centrifugation at 4000 g and 4 °C for 15 min. We washed the pelleted cells using 1 mL of cold PBS and resuspended them in ~1.8 mL of cold PBS. Subsequently, we transferred 90 μL of the suspended cells into a 96-well PCR plate,and used a thermocycler to heat the resulting cell suspension for 5 min in a temperature range of 55-95 °C (55.0, 58.6, 62.6, 67.6, 72.5, 77.5, 82.4, 87.4, 91.4, 93.9, and 95.0 °C), followed by a 30-second incubation at 4 °C. Immediately after the incubation, we transferred 80 μL of each cell suspension to a 96-well microplate, and used a Tecan Spark plate reader ($\lambda_{ex}$ = 400 nm, $\lambda_{em}$ = 512 nm) to measure its fluorescence intensity. We used the unheated cell suspension and the GFP-free cell suspension (cells without a GFP-coding gene) as positive and blank controls, respectively. We followed the same procedure as described above to calculate the residual fluorescence, to fit the resulting data to a sigmoidal model, and to derive the midpoint temperature of the transition curve ($T_m$) for each mutant, as well as for ancestral GFP (WT) (see Fig. 2e).

### Protein purification and melting temperature ($T_m$) determination by circular dichroism

We purified fluorescent proteins for ancestral GFP and its variants and measured their melting temperatures ($T_m$) using circular dichroism. To purify the proteins, we inoculated 7.5 μL of a glycerol stock of ancestral GFP or its variants into a 250 mL flask containing 25 mL of LB supplemented with 50 μg/mL Kanamycin. We incubated the culture at 37 °C with shaking at 220 rpm for 12 h. We pelleted the cells by centrifuging the overnight culture at 3000 g and 4 °C for 10 min. We resuspended the cells in 50 mL of LB (containing 50 μg/ mL Kanamycin and 0.2% (w/v) arabinose) and continued the incubation at 37 °C with shaking at 250 rpm for another 12 h. We collected the cells by centrifuging the overnight culture at 3,000 g and 4 °C for 15 min, and washed the pelleted cells once with cold 2 × PBS. Then, we pelleted the cells again at 3,000 g and 4 °C for 15 min and resuspended them in 50 mL of 2 × PBS. We broke the cells to release the fluorescent proteins using a pressure cell cracker (UH-06, Union-biotech) and pelleted the insoluble fractions by centrifugation twice at 18,000×g for 10 min. We filtered the resulting supernatant with a 0.45 μM filter and used the high-affinity Ni-charged resin FF (L00666, GenScript) for purification, following the manufacturer's protocol. We used PD-10 desalting columns (52-1308-00 BB, GE Healthcare) to desalt the purified proteins according to the manufacturer's instructions, and concentrated the desalted fluorescent proteins using an Amicon Ultra-15 filter (UFC901008, Merck). We assessed the purity of the concentrated proteins using SDS-PAGE and quantified the concentration using a Bradford Protein Assay Kit (P0006, Beyotime).

We diluted the purified proteins to a concentration of 0.1–0.5 mg/ mL using MiliQ water. We then transferred 200 μL of diluted proteins

into a 1 mm thick cuvette and placed it in a Circular Dichroism spectropolarimeter (Chirascan V100, Applied Photophysics Ltd). We heated these samples from 50 to 90 °C at a rate of 1 °C per minute and measured their absorbance from 195 nm to 255 nm. We used the Global 3 V1.2 (Applied Photophysics Ltd) to calculate the melting temperature for each sample.

### Cell-free expression of stabilizing mutants at different temperatures and determination of solubility by ELISA

To exclude the possibility that our observations at high temperature are caused by heat shock proteins, we performed new experiments, in which we expressed our proteins in a cell-free expression system (PURExpress In Vitro Protein Synthesis Kit, E6800S, NEB) at low, medium and high temperatures. This system contains only the components necessary for in vitro transcription and translation, all purified from *E. coli*, and completely avoids interference by heat shock proteins. Specifically, we replaced the DHFR gene in the PURExpress control DHFR plasmid with each of the genes of the full-length stabilizing mutants using a seamless cloning Kit (D7010M, Beyotime). We transformed the resulting plasmids into *E. coli* BW27783 by heat shock transformation and isolated the correctly constructed plasmids as templates for performing the in vitro protein synthesis reaction. The protein synthesis reaction contained 2 μL solution A, 1.5 μL solution B, 0.5 μL RNase inhibitor (8 U/μL; M0314S, NEB) and 1.0 μL template (50 ng/μL). The mixed solution was incubated in a PCR thermocycler tube at 25 °C and 44 °C for 8 h and 37 °C for 4 h. We then followed the manufacture's protocol to quantify the amount of soluble fluorescent proteins in each solution by using a GFP ELISA kit (ab171581, Abcam), which can detect GFP from *Aequorea Victoria* and its enhanced and superfold variants.

### Single-molecule real-time sequencing

As described in a previous study[64], we barcoded GFP variants of each replicate population for SMRT (Pacific Biosciences, PacBio) sequencing by two-step PCRs. We used Q5 High-Fidelity DNA Polymerase (M0491L, NEB) to minimize the incidence of mutations during the PCR amplifications. Specifically, we first performed a 12-cycle PCR using primers tsmrtF/tsmrtR (Table S4) to amplify GFP variants for each replicate population. The PCR reaction contained 15.7 μL ddH$_2$O, 6 μL 5×Q5 reaction buffer, 6 μL 5×Q5 high GC enhancer, 0.6 μL dNTPs (10 nM), 0.2 μL each primer (10 μM), 0.3 μL Q5 High-Fidelity DNA Polymerase, and 1 μL template (1 ng/μL). We performed the PCR with the following thermocycling program: 98 °C/30 s; 14 cycles of 98 °C/10 s, 60 °C/10 s, and 72 °C/25 s; 72 °C/2 min. We added 0.5 μL *Dpn*I, 0.5 μL Exonuclease I (EN0581, Fermentas), as well as 0.5 μL ddH$_2$O to the PCR product, and incubated the mixture at 37 °C for 1 h to digest the template plasmid and the primers. We then incubated the mixture at 80 °C for 20 min to inactivate the enzymes. Subsequently, we used 1 μL of PCR product as a template for a barcoding PCR in a 50 μL volume, which utilized a unique combination of a forward and a reverse barcode-tagged primers (Table S4) to barcode each replicate population. Each barcode-tagged primer has a unique 16-bp DNA sequence. The barcoding PCR mix contained 25.5 μL ddH$_2$O, 10 μL 5×Q5 reaction buffer, 10 μL 5×Q5 high GC enhancer, 1 μL dNTPs (10 mM), 1 μL each primer (10 μM), 0.5 μL Q5 High-Fidelity DNA Polymerase, and 1 μL template. We executed the PCR reaction with the following program: 98 °C/30 s; 25 cycles of 98 °C/15 s, 68 °C/10 s and 72 °C/25 s; 72 °C/5 min. We purified the barcode-tagged PCR products by using a QIAquick PCR purification kit, and estimated their quality and concentration with agarose gel electrophoresis, a UV-Vis spectrophotometer (NanoDrop, Thermo Fisher Scientific), and a Qubit 4 Fluorometer (Invitrogen). We also used the ancestral GFP gene as a template for preparing barcode-tagged PCR products by following the same procedure to estimate the incidence of any errors that might

have occurred during library preparation and sequencing. We pooled 100 ng DNA for each replicate population and for every generation in a single tube, and then gel-purified the mixture by using a QIAquick Gel Extraction Kit (Qiagen). We sent ~1 μg of the resulting amplicons to the Functional Genomics Center Zurich for sequencing with the PacBio Sequel II platform.

### Primary data analysis

We used the SMRT Link V9.0.0.92188 software package (PacBio) to perform the primary sequencing data analysis. Specifically, we assembled consensus reads from single-stranded subreads by using the protocol "Circular Consensus Sequences (CCS)" and set the insert GFP length to 720–1200 bp, the full-pass subread number to ≥3, and the predicted consensus accuracy to ≥0.99. To demultiplex the resulting sequencing data by using the "Demultiplex Barcodes" application, we set the "Minimum Barcode Score" to 80 and the "Filter Minimum Barcode Quality" to 26. We used the 'Mapping' application to map the demultiplexed reads to the ancestral GFP sequence by setting the minimum mapped length to 700 bp and the minimum mapped concordance to 70%. We then selected the mapped reads that had an average Phred quality above 20, covered the entire GFP coding region, and didn't have indels, which yielded an average of 2347 reads for each replicate population at each generation (Table S3). We used the resulting reads for all further analyses.

### Identifying mutations and mutation combinations

Our analysis focused on SNPs (point mutations) because > 90% of sequencing errors during SMRT sequencing come from indels[66], and because most indels render fluorescent proteins non-fluorescent. We treated a SNP as a true-positive only if the Phred quality score of a mismatch of a variant sequence to the GFP reference sequence was above 20. We wrote Python scripts (Python 3.10.4) to search point mutations, and their combinations and calculated their frequencies in each replicate population.

### Statistical analyses

Unless specified otherwise, we performed one-way ANOVA with Dunett's post hoc test for statistical data analysis by using R version 4.2.1.

### Reporting summary

Further information on research design is available in the Nature Portfolio Reporting Summary linked to this article.

## Data availability

All data are available in the manuscript or the supplementary materials. All sequencing data is available at GenBank under accession number KIDR00000000. Source data are provided as a Source Data file. Source data are provided with this paper.

## Code availability

Custom code used in this study is available at Zenodo under the accession number https://doi.org/10.5281/zenodo.10146677 [https://zenodo.org/records/10146677].

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

## Acknowledgements

We acknowledge the experimental supports of the flow cytometry core facility and the instrumentation and service center for molecular sciences at Westlake University, and the flow cytometry facility and the functional genomics center at the University of Zurich. We thank Prof. Ren Sun and Dr. Boxiao Wang for helpful discussions and suggestions. We would like to acknowledge support by Westlake Education Foundation (J.Z.), the URPP Evolution in Action (J.Z.), the National Natural Science Foundation of China grant 32270669 (J.Z.), the European Research Council under grant agreement 739874 (A.W.), and the Swiss National Science Foundation grant 31003A_172887 (A.W.).

## Author contributions

J.Z. designed the experiments. J.Z., N.G., Y.H. and X.G. performed the experiments. N.G., J.Z. and A.W. contributed to data analysis. J.Z. and A.W. wrote the paper. All authors read and edited the paper.

## Competing interests

The authors declare no competing interests.
