## [Peer Review File · Nature Communications]

REVIEWER COMMENTS

Reviewer #1 (Remarks to the Author):

Zheng et al investigate the effects of temperature on the laboratory evolution of fluorescent proteins towards yellow fluorescence. They use an experimental system that has been very productive in the past for testing basic predictions about the nature of evolutionary change. The paper finds that under their experimental conditions, high temperature leads to slower evolution of yellow fluorescence. While I find this study in principle interesting, I have several questions about the work that the authors should address before I can recommend publication in Nature Communications.

First, I was a little confused by the introduction. Excess thermal stability is a well-known promoter of protein evolvability, especially in the field of protein engineering (for example PMID 16581913). Several dedicated tools for producing such excess stability have been produced specifically for this purpose (one recent example that makes this argument is here PMID 36452862). The importance of a stability buffer in natural evolution has also been demonstrated before (for example see PMID 23682315). None of this work is cited, instead a rather general introduction is given, discussing the general effects of temperature on life. My feeling was that this creates a rather artificial strawman (though a beautifully presented one), which serves to create an expectation (that high temperature accelerates protein evolution) that I think no protein engineer would share. The authors of course have the freedom to motivate their work whichever way they see fit, but I think they may consider also presenting arguments for the alternative view (that high temperature makes it harder to evolve proteins that are not already adapted to that temperature). This of course comes at the expense of the findings perhaps seeming a little less surprising.

Another, more fundamental concern for me is the experimental design. What the authors want is to quantify how temperature affects the evolution of one protein, separating this as much of possible from effects on expression and other cellular processes. But the high temperature setting is firmly in heat shock territory for e.coli. Cells will synthesize large quantities of heat shock proteins. Translation rates are likely altered and so on. The authors I guess hope to deal with this by normalizing for total FP amount. But later in the paper, they make several statements about mutations changing the amount of total protein. They interpret this as stemming from increased thermal stability. I find this extremely implausible. All experiments are done far below the melting point of the variants of interest. In one case, the T_m is increased by less than 5 C, yet the amount of soluble protein is increased more than 70fold at 44C. I consider it essentially impossible that the effect on thermal stability explains this difference. Instead, I think it is quite likely that what the authors term 'stability increasing' mutations exert their effects by changing transcription or translation rates. I think this problem may also affect what variants survive cell sorting. Total protein amounts can only be controlled for after sorting. I therefore think it likely that stabilizing mutations really help the protein increase its translation or transcription rates in the face of a highly stressed cytoplasm. The point is that the situation might be quite different if the same experiment were performed in a strain that was already adapted to high temperatures (and therefore does not trigger a heatshock response at these temperatures).

One last small technical point: Fold changes should be shown on log scales in the figures. The current figures emphasize increases, but compress decreases between 0 and 1.

Lastly, I think the authors may consider toning down their discussion. It seems unreasonable to me to extrapolate from the laboratory evolution of a fluorescent protein to organismal responses to climate change.

Reviewer #2 (Remarks to the Author):

This study by Zheng et al. addresses the fundamental question how temperature influences evolvability. They use a simple model system, namely the directed evolution of green fluorescent protein (GFP) towards a yellow fluorescence phenotype. Random mutations are generated by error-prone PCR and the libraries are expressed in *E. coli*. Selection for fitness (= yellow fluorescence) is done at three different temperatures (25, 37, and 44 °C) by fluorescence-activated cell sorting (FACS) and evolutionary dynamics are evaluated by single-molecule real-time sequencing. The simplicity of the system is important to reduce the effects of interfering cellular processes on the evolutionary outcome. The hypotheses were tested thoroughly using adequate statistics. It was found that "neofunctionalizing" mutations giving rise to yellow fluorescence appeared faster at lower temperature, where their destabilizing effects on protein structure can be tolerated. At higher temperature, stabilizing mutations accumulated first and more rounds of evolution were needed to obtain the desired phenotype. This is a very interesting finding, however, it remains unclear to what extent the general conclusion that low temperature accelerates adaptive evolution holds true beyond this very simple model system. The statements on the evolutionary rescue of species endangered by climate change should certainly be removed. This work should only be published, if the following points have been addressed adequately:

- 1) The protein sequence of the "ancestral GFP" used here as the starting point should be given, as the numbering and mutations seems odd. The well-known key mutations that shift GFP's fluorescence from green to yellow are T203Y/S65G, with tyrosine in position 203 pi-stacking on the chromophore. I assume the neofunctionalizing mutations C204Y and S66G found here are the exact same positions and the numbering is just off by one? Also, why is there a cysteine in the starting sequence in that crucial position?
- 2) Seminal literature on GFP/YFP should be cited. The authors make it sound as if these key mutations were unknown. When Tsien and Remington reported the first GFP structure in 1996, they already mentioned that aromatic residues in position 203 lead to red-shifted excitation and emission. More detailed studies followed. All identified mutations should also be mapped on a GFP crystal structure.
- 3) Did the other aromatic amino acids never come up in position 203(204) as neofunctionalizing mutations, especially at the lower temperature? They should also have a clear spectral effect.
- 4) The GFP variants are only characterized in lysate. It would be better to work with purified proteins for more precise protein stability measurements. The destabilizing effect of a mutation measured with CD melting curves should be compared to the fluorescence readout.
- 5) Can the finding that low temperature accelerates adaptive evolution really be generalized? What about the evolution of an essential gene/protein? For instance, there is a wealth of literature on the evolution of antibiotic resistances. Is there anything known on temperature dependence that could be discussed in this context?
- 6) I was wondering if the advice for protein engineers could be even a bit more specific? How about: "If you want to evolve a protein for a new function, make sure you evolve at a temperature range way below its T_m ." In the end "high" and "low" temperature only makes sense with a proper reference.
- 7) The statements on the evolutionary rescue of species endangered by climate change should be removed from abstract and discussion. The results obtained here do not seem transferable to the survival of higher organisms.
- 8) In Figure 3d, the "L, M, H" labels seem incorrectly placed. They are not matching the color-coding of the rest of the manuscript (L=blue, M=grey, H=red).

9) In Figure 4d, the labels are hard to read, I would advise placing them in a different way. In panel e, is it the relative specific yellow fluorescence of only the "L" populations?

10) The manuscript title must be updated in the Supporting Information. It says: "Lowered temperature accelerates Darwinian evolution" as supposed to "High temperature delays and low temperature accelerates evolution of a new protein phenotype"

RESPONSE TO REVIEWERS' COMMENTS

Reviewer #1 (Remarks to the Author):

Zheng et al investigate the effects of temperature on the laboratory evolution of fluorescent proteins towards yellow fluorescence. They use an experimental system that has been very productive in the past for testing basic predictions about the nature of evolutionary change. The paper finds that under their experimental conditions, high temperature leads to slower evolution of yellow fluorescence. While I find this study in principle interesting, I have several questions about the work that the authors should address before I can recommend publication in Nature Communications.

Reply: Before our point-by-point reply, we would like to thank you for your constructive comments and suggestions. They prompted new experiments/analyses and more discussions, which helped us to greatly improve the quality of the manuscript. We have carefully revised the manuscript to address all concerns raised. Please find our point-by-point reply to your suggestions below! All changes made in the revised manuscript and the supplementary materials are highlighted in green.

1. First, I was a little confused by the introduction. Excess thermal stability is a well-known promoter of protein evolvability, especially in the field of protein engineering (for example PMID 16581913). Several dedicated tools for producing such excess stability have been produced specifically for this purpose (one recent example that makes this argument is here PMID 36452862). The importance of a stability buffer in natural evolution has also been demonstrated before (for example see PMID 23682315). None of this work is cited, instead a rather general introduction is given, discussing the general effects of temperature on life. My feeling was that this creates a rather artificial strawman (though a beautifully presented one), which serves to create an expectation (that high temperature accelerates protein evolution) that I think no protein engineer would share. The

authors of course have the freedom to motivate their work whichever way they see fit, but I think they may consider also presenting arguments for the alternative view (that high temperature makes it harder to evolve proteins that are not already adapted to that temperature). This of course comes at the expense of the findings perhaps seeming a little less surprising.

Reply: Thank you for this suggestion. In response, we have revised the abstract and introduction to do exactly what you requested. That is, we now discuss how high temperature might promote protein evolvability, and the alternative view that it might hinder protein evolvability by destabilizing proteins. We also cited the literature you mention in the revised manuscript (refs. 31-33,51 and 52). Specifically, lines 18-23 on pages 2 read:

On the one hand, high temperature may accelerate phenotypic evolution because it accelerates most biological processes. On the other hand, it may also slow phenotypic evolution, because proteins are usually less stable at high temperatures and therefore less evolvable. To validate these conflicting hypotheses experimentally, we evolved a green fluorescent protein in *E. coli* towards the new phenotype of yellow fluorescence at different temperatures.

and lines 56-61 on page 4 read:

Because increasing temperature accelerates most processes in the biosphere^{1,13,30}, it may also accelerate the evolution of new phenotypes. Alternatively, increasing temperature may impede phenotypic evolution, because high temperature can destabilize proteins and thus reduce protein evolvability³¹⁻³³. We validated these conflicting hypotheses by experimentally evolving green fluorescent protein (GFP, a derivative of the green fluorescent protein of the jellyfish *Aequorea Victoria* (avGFP); see Fig. S1)³⁴ in *E. coli*.

as well as line 423 on page 27 reads:

Protein stability is crucial for protein evolution^{31,49,50} and is a major goal in protein engineering^{51,52}.

2. Another, more fundamental concern for me is the experimental design. What the authors want is to quantify how temperature affects the evolution of one protein, separating this as much of possible from effects on expression and other cellular processes. But the high temperature setting is firmly in heat shock territory for e.coli. Cells will synthesize large quantities of heat shock proteins. Translation rates are likely altered and so on. The authors I guess hope to deal with this by normalizing for total FP amount. But later in the paper, they make several statements about mutations changing the amount of total protein. They interpret this as stemming from increased thermal stability. I find this extremely implausible. All experiments are done far below the melting point of the variants of interest. In one case, the T_m is increased by less than 5 C, yet the amount of soluble protein is increased more than 70fold at 44C. I consider it essentially impossible that the effect on thermal stability explains this difference. Instead, I think it is quite likely that what the authors term ‘stability increasing’ mutations exert their effects by changing transcription or translation rates. I think this problem may also affect what variants survive cell sorting. Total protein amounts can only be controlled for after sorting. I therefore think it likely that stabilizing mutations really help the protein increase its translation or transcription rates in the face of a highly stressed cytoplasm. The point is that the situation might be quite different if the same experiment were performed in a strain that was already adapted to high temperatures (and therefore does not trigger a heatshock response at these temperatures).

Reply: Thank you for raising this important concern, which we addressed with new experiments. Unfortunately, an experiment with a strain adapted to high temperature is not feasible, because obtaining such a strain might itself require years of experimental evolution, after which we would still have to repeat all the evolution experiments that we performed in our study with it. However, we performed two complementary experiments that are equally informative and that support our original conclusion.

Stabilizing mutations may indeed increase the total amount of soluble fluorescent proteins by improving transcription, translation, protein folding or thermodynamic stability. The first two phenomena can occur when a mutation increases expression yields by enhancing mRNA stability, or when it reduces protein misfolding and insolubilization by slowing translation (1, 2). If this is the main contributor to the greater improvement in solubility caused by stabilizing mutations at high temperature, we would expect that 1) different codons encoding the same stabilizing amino acid substitution lead to substantial variation in solubility (and thus fluorescence), and 2) such solubility variation should be greater at high temperature than at medium and low temperatures. To find out whether this is case, we engineered all possible synonymous mutations into the codon encoding each of the three stabilizing mutations F65L, V164A, and I172V.

We then expressed all these variants at low, medium and high temperatures, and quantified their fluorescence. Because the synonymous variants encode the same amino acids, the proteins expressed from them will have the same specific fluorescence intensity. We can thus directly quantify their solubility by measuring their fluorescence. The results refute the hypotheses that changes in transcription or translation are at the root of our observations. Specifically, solubility varies little between synonymous variants for all stabilizing mutants. Even more importantly, the extent of variation in solubility is similar between low, medium, and high temperatures (see Fig. S7). These observations suggest that stabilizing mutations do not improve solubility at high temperature mainly by enhancing transcription or translation. We now discuss this new data in the revised manuscript on lines 209-215 (page 14), which read:

One possible explanation is that these stabilizing mutations increase expression yields by enhancing the transcription and stability of mRNA, or by reducing protein misfolding and insolubilization through slowing translation^{33,45}. To test this hypothesis, we engineered all possible synonymous mutations into the codon encoding each of

the three stabilizing mutations. The resulting data shows that none of these synonymous mutations substantially changes green fluorescence at either low, medium or high temperature for any of these variants (Fig. S7). Thus, the greater benefit of these mutations at higher temperatures does probably not just result from their effects on transcription and translation.

If stabilizing mutations do not act primarily by affecting transcription and translation, they need to act by enhancing protein folding and/or thermostability. However, as the reviewer correctly states, high temperatures cause high expression of heat shock proteins (mainly molecular chaperones and proteases) which can also promote protein folding or degradation (3). To exclude the possibility that our observations at high temperature are caused by heat shock proteins, we performed new experiments, in which we expressed our proteins in a cell-free expression system (PURExpress In Vitro Protein Synthesis Kit, NEB) at low, medium and high temperatures. This system only contains the necessary components for in vitro transcription and translation, all purified from *E. coli*, and completely avoids interference by heat shock proteins. The resulting in vitro data support our in vivo data. Specifically, the three stabilizing mutations increase solubility on average 10.3-fold at high temperature, but only 2.4-fold at medium temperature and 1.9-fold at low temperature (see the new Fig. S8). The difference in magnitude between the in vivo and in vitro solubility improvement might be due to the fact that the in vivo environment contains chaperones and proteases, which can affect the folding and degradation of different variants to a different extent (4).

In sum, our in vitro expression experiment excludes the possibility that high expression of heat shock proteins at high temperatures drives our observations. Therefore, our conclusions should still hold if we could perform our experiments in a strain that was already adapted to high temperatures.

We now present our new data in fig. S8, and discuss these observations on lines 222-228 (pages 14 and 15), which read:

We observed a similar solubility increase in a cell-free expression system that only contained the components necessary for in vitro transcription and translation. Specifically, in this cell-free system all three stabilizing mutations increase solubility on average 10.3-fold at high temperature, but only 2.4-fold at medium temperature and 1.9-fold at low temperature (Fig. S8). The difference in magnitude between the in vivo and in vitro solubility improvement might be due to the fact that the in vivo environment contains chaperones and proteases, which can affect the folding and degradation of different variants to a different extent ⁴⁷.

Finally, we would like to reply to the reviewer's question why a stabilizing mutation that increases the already high melting temperature ($T_m > 70$ °C, Fig. 2e) by a few degrees can increase the amount of soluble fluorescent proteins at 44 °C many-fold, since our proteins should already be "stable enough" at this temperature. This is possible, because elevated temperature can interfere with the post-translational maturation of GFP rather than causing inactivation of the mature protein (5). The most temperature-sensitive step for post-translational maturation is the correct folding of apoprotein into a catalytically active conformation, which is essential for the maturation of the chromophore. Elevated temperatures can lead to improper folding of the apoprotein and thus reduce the yield of soluble fluorescent proteins. This folding defect is a common phenomenon during GFP expression in vivo and can be rescued by stabilizing mutations. For example, the two mutations V163A and S175G can increase the soluble expression of avGFP (the green fluorescent protein of the jellyfish *Aequorea Victoria*, the ancestor of our GFP) at 37 °C by almost 5-fold (5). We observed one of these stabilizing mutation (V163A in avGFP, V164A in our GFP) in our study. Stabilizing mutations may thus be even more important for improving the solubility of

fluorescent proteins at our higher temperature of 44 °C. To avoid adding potentially distracting material, we did not add the above explanation to the revised manuscript.

3. One last small technical point: Fold changes should be shown on log scales in the figures. The current figures emphasize increases, but compress decreases between 0 and 1.

Reply: Good point! Now we plot the data on a logarithmic scale whenever appropriate, including in Figs 1b, 2b, 3a, 3b, 5b, 5d, 5g, 5h, S10 and S11.

4. Lastly, I think the authors may consider toning down their discussion. It seems unreasonable to me to extrapolate from the laboratory evolution of a fluorescent protein to organismal responses to climate change.

Reply: Point well taken. In the revised, shortened, and toned-down discussion, we tried to balance this concern with our intention to communicate the broader significance of temperature for adaptive evolution. Any further suggestions are welcome! Specifically, lines 457-465 on pages 28 and 29 in the revised manuscript now read:

Another implication relates to ongoing global climate change, which can exert severe thermal stress on organisms and force them to undergo adaptive evolution to higher temperatures^{61,62}. Although global climate change itself alters the average global environment only very gradually, one of its consequences is an increase of extreme climatic events, such as extreme heat waves with rapid temperature increase^{49,62-64}. Such a rapid temperature increase can purge neofunctionalizing mutations that destabilize proteins. Consequently, it may impede phenotypic evolution, and thus result in decreased genetic diversity and biodiversity. Most generally, our work suggests that a temperature change can profoundly affect how life finds a way towards phenotypic innovation.

Response to reviewer #2

Reviewer #2 (Remarks to the Author):

This study by Zheng et al. addresses the fundamental question how temperature influences evolvability. They use a simple model system, namely the directed evolution of green fluorescent protein (GFP) towards a yellow fluorescence phenotype. Random mutations are generated by error-prone PCR and the libraries are expressed in E. coli. Selection for fitness (= yellow fluorescence) is done at three different temperatures (25, 37, and 44 °C) by fluorescence-activated cell sorting (FACS) and evolutionary dynamics are evaluated by single-molecule real-time sequencing. The simplicity of the system is important to reduce the effects of interfering cellular processes on the evolutionary outcome. The hypotheses were tested thoroughly using adequate statistics. It was found that “neofunctionalizing” mutations giving rise to yellow fluorescence appeared faster at lower temperature, where their destabilizing effects on protein structure can be tolerated. At higher temperature, stabilizing mutations accumulated first and more rounds of evolution were needed to obtain the desired phenotype. This is a very interesting finding, however, it remains unclear to what extent the general conclusion that low temperature accelerates adaptive evolution holds true beyond this very simple model system. The statements on the evolutionary rescue of species endangered by climate change should certainly be removed. This work should only be published, if the following point have been addressed adequately:

Reply: We appreciate your positive evaluation of our work and your helpful comments and suggestions, which helped us to improve the quality of the manuscript and make it more readable. We have thoroughly revised the manuscripts according to your feedback and highlighted all changes in green. Please find our point-by-point responses below, and please see our reply to point 7 below for our revisions with regard to statements about evolutionary rescue.

1) The protein sequence of the “ancestral GFP” used here as the starting point should be given, as the numbering and mutations seems odd. The well-known key mutations that shift GFP’s fluorescence from green to yellow are T203Y/S65G,

with tyrosine in position 203 pi-stacking on the chromophore. I assume the neofunctionalizing mutations C204Y and S66G found here are the exact same positions and the numbering is just off by one? Also, why is there a cysteine in the starting sequence in that crucial position?

Reply: You're perfectly correct, and we apologize for the oversight of not explaining the reason for this shifted coordinate system. Our ancestral GFP is a derivative of the *Aequorea victoria* green-fluorescent protein (avGFP). Compared to avGFP, our GFP has six amino acid substitutions and a methionine insertion at the second residue, which causes the positions of the same mutations to differ by one. Thus, the positions of C204Y and S66G in our GFP are the same as T203Y and S65G in avGFP. A cysteine at residue 204 in our GFP (residue 203 in avGFP) can cause a green shift of the emission peak, as proven in a previous study (6). To avoid confusion, we have added a new figure, Fig. S1, in the revised manuscript, showing the differences in protein sequence between our GFP and avGFP. In addition, we now also discuss these differences in lines 59-61 on page 4, lines 150-153 on page 10, lines 193-195 on page 13 and lines 472-474 on page 29.

Lines 59-61 on page 4 read:

We validated these conflicting hypotheses by experimentally evolving green fluorescent protein (GFP, a derivative of the green fluorescent protein of the jellyfish *Aequorea Victoria* (avGFP); see Fig. S1)³⁴ in *E. coli*.

Lines 150-153 on page 10 read:

These data are also supported by previous observations, in which the same four mutations changed excitation and/or emission spectra in the genetic background s of avGFP (mutations 65G, 167T, 167V and 203C in avGFP³⁶⁻³⁸; see Fig. S1 for the differences in the coordinate system between our GFP and avGFP).

Lines 193-195 on page 13 read:

Previous studies had revealed that these three mutations can improve the folding stability of YFP⁴³ and of avGFP (mutations 64L, 163A and 171V in the coordinate system of avGFP, see also Fig. S1)^{38,44}.

And lines 471-473 on page 29 read:

We refer to the encoded GFP protein as our 'ancestral' GFP. **It carries a methionine insertion at the second residue and six amino acid substitutions compared to the *Aequorea victoria* green-fluorescent protein (avGFP) (Fig. S1).**

2) Seminal literature on GFP/YFP should be cited. The authors make it sound as if these key mutations were unknown. When Tsien and Remington reported the first GFP structure in 1996, they already mentioned that aromatic residues in position 203 lead to red-shifted excitation and emission. More detailed studies followed. All identified mutations should also be mapped on a GFP crystal structure.

Reply: Thank you for highlighting these oversights. We made the requested changes. First, we mapped all these seven key mutations onto the GFP structure in the new Fig. S3. Second, we also added relevant information/discussions and cited pertinent literature on lines 59-61 of page 4, lines 150-153 of page 10 and lines 193-195 of page 13, as also indicated in our last response to point 1.

3) Did the other aromatic amino acids never come up in position 203(204) as neofunctionalizing mutations, especially at the lower temperature? They should also have a clear spectral effect.

Reply: To follow up on this concern, we have carefully checked our high throughput sequencing data, and did not find any other mutations towards aromatic amino acids at this position that reached the threshold frequency of 5%. We suspect that mutations at other positions prevent their occurrence through negative epistatic interactions.

4) The GFP variants are only characterized in lysate. It would be better to work with purified proteins for more precise protein stability measurements. The

destabilizing effect of a mutation measured with CD melting curves should be compared to the fluorescence readout.

Reply: In response to this concern, we purified the proteins and measured their melting temperatures by CD. We also measured the in vivo melting temperatures of all variant and ancestral GFPs by using cells that expressed the corresponding fluorescent proteins. We compared these results with those from cell lysates and found an identical pattern. In particular, all four neofunctionalizing mutations decreased the melting temperature of GFP, but all stabilizing mutations increased it.

In the revised manuscript, we replaced the T_m data measured by using cell lysates with the newly measured data of in vivo melting temperatures in Fig. 2e. The T_m data for using cells lysates and purified fluorescent proteins are shown in Fig. S6a and b, respectively. We added this information in lines 193-200 on page 13, which read:

Previous studies had revealed that these three mutations can improve the folding stability of YFP⁴³ and of avGFP (mutations 64L, 163A and 171V in the coordinate system, see also Fig. S1)^{38,44}. Absent other phenotypes, we hypothesized that these mutations are beneficial, because they can also increase protein folding stability in the genetic background of our ancestral GFP. To test this hypothesis, we engineered each of the three mutations into ancestral GFP, and measured the melting temperature T_m of the engineered fluorescent protein. Indeed, all three mutations significantly increased this melting temperature (One-way ANOVA with Dunnett's post hoc test, $P < 0.001$; Fig. 2e and Fig. S6).

5) Can the finding that low temperature accelerates adaptive evolution really be generalized? What about the evolution of an essential gene/protein? For instance, there is a wealth of literature on the evolution of antibiotic resistances. Is there anything known on temperature dependence that could be discussed in this context?

Reply: We're sorry for not making this clearer in our original discussion. In short, we believe that our findings would not depend on whether a protein is essential or not. They should apply to the adaptive evolution of functional proteins whenever neofunctionalizing mutations destabilize proteins. However, whether our observations still hold for proteins in which neofunctionalizing mutations are not destabilizing remains unknown. To make this clearer, we revised this part in lines 434-444 on pages 27 and 28, which now read:

At any one location on Earth, temperature may rapidly fluctuate by many degrees due to diurnal cycles and seasonal changes. Such temperature changes can alter the strength of selection to modify protein stability during the evolution of new protein functions. Our experiments show that an increase in temperature strengthens and a decrease weakens selection for stability (Fig. 3d). Because adaptive mutations that bring forth new protein functions usually destabilize proteins^{40-42,53}, stronger selection on protein stability at high temperatures can delay the spreading of neofunctionalizing mutations through an evolving population. Consequently, increasing temperature can slow down and lowering temperature can speed up the evolution of a new phenotype (Fig. 1). We expect that our observations will apply whenever neofunctionalizing mutations reduce the stability of adaptively evolving proteins. Whether they also apply when neofunctionalizing mutations do not reduce stability³⁵ is an exciting question for future work.

6) I was wondering if the advice for protein engineers could be even a bit more specific? How about: "If you want to evolve a protein for a new function, make

sure you evolve at a temperature range way below its T_m .” In the end “high” and “low” temperature only makes sense with a proper reference.

Reply: Done! We have revised the pertinent passage on lines 453-457 of page 28 accordingly, which now reads:

Our work has several practical implications. For example, it suggests that lowering temperature substantially below that required for soluble protein expression can help evolve or engineer proteins with new functions, because it helps function-altering mutations to manifest their beneficial effects. This strategy may be especially helpful when conventional approaches fail, because neofunctionalizing mutations are too strongly destabilizing^{42,59,60}.

7) The statements on the evolutionary rescue of species endangered by climate change should be removed from abstract and discussion. The results obtained here do not seem transferable to the survival of higher organisms.

Reply: We agree that we needed to tone-down this part of the discussion. At the same time, we respectfully submit that our observations on protein stability are relevant for the survival of whole organisms and their evolutionary adaptation to high temperature. For example, in a recent study (Toll-Riera et al., Science Advances 2022), we showed that cold-loving bacteria experience a hard limit on evolutionary adaptation to temperatures beyond 30 degrees. That is, they go extinct beyond 30 °C and evolutionary rescue becomes impossible, because their chaperones can neither handle the many misfolded proteins at higher temperatures, nor can they evolve sufficiently rapidly to cope with this misfolding load. Nevertheless, in response to this concern, we toned down abstract and discussion to read (lines 31-33 on page 2 and lines 457-465 on pages 28 and 29):

Our observations have broad implications for the evolutionary engineering of improved proteins, and for our understanding of how temperature changes affect evolutionary

adaptations and innovations.

and

Another implication relates to ongoing global climate change, which can exert severe thermal stress on organisms and force them to undergo adaptive evolution to higher temperatures^{61,62}. Although global climate change itself alters the average global environment only very gradually, one of its consequences is an increase of extreme climatic events, such as extreme heat waves with rapid temperature increase^{49,62–64}. Such a rapid temperature increase can purge neofunctionalizing mutations that destabilize proteins. Consequently, it may impede phenotypic evolution, and thus result in decreased genetic diversity and biodiversity. Most generally, our work suggests that a temperature change can profoundly affect how life finds a way towards phenotypic innovation.

8) In Figure 3d, the "L, M, H" labels seem incorrectly placed. They are not matching the color-coding of the rest of the manuscript (L=blue, M=grey, H=red).

Reply: Thank you for pointing out this mistake. We have corrected it now in the revised Fig. 3d.

9) In Figure 4d, the labels are hard to read, I would advise placing them in a different way. In panel e, is it the relative specific yellow fluorescence of only the "L" populations?

Reply: For fig. 4d, we now changed the positions of these labels to make them easier to read. Fig. 4e indeed refers only to the relative specific yellow fluorescence for these three double mutants at low temperature, as is now indicated in the caption (lines 309 and 310). We observed a similar pattern at medium and high temperatures and thus didn't show the redundant data.

10) The manuscript title must be updated in the Supporting Information. It says: "Lowered temperature accelerates Darwinian evolution" as supposed to "High temperature delays and low temperature accelerates evolution of a new protein

phenotype”

Reply: Done.

Reference

1. H. M. Strobel, E. K. Horwitz, J. R. Meyer, Viral protein instability enhances host-range evolvability. *PLOS Genet.* **18**, e1010030 (2022).
2. G. L. Rosano, E. A. Ceccarelli, Rare codon content affects the solubility of recombinant proteins in a codon bias-adjusted Escherichia coli strain. *Microb. Cell Fact.* **8**, 1–9 (2009).
3. F. Arsène, T. Tomoyasu, B. Bukau, The heat shock response of Escherichia coli. *Int. J. Food Microbiol.* **55**, 3–9 (2000).
4. N. Tokuriki, D. S. Tawfik, Chaperonin overexpression promotes genetic variation and enzyme evolution. *Nat. 2009 4597247.* **459**, 668–673 (2009).
5. K. R. Siemering, R. Golbik, R. Sever, J. Haseloff, Mutations that suppress the thermosensitivity of green fluorescent protein. *Curr. Biol.* **6**, 1653–1663 (1996).
6. J. Zheng, N. Guo, A. Wagner, Selection enhances protein evolvability by increasing mutational robustness and foldability. *Science.* **370**, eabb5962 (2020).

REVIEWERS' COMMENTS

Reviewer #1 (Remarks to the Author):

The authors added some convincing in vitro translation assays that I think significantly strengthen their conclusions.

My only remaining concern is that they give a sensible explanation about how stabilizing mutations probably exert their effects not by raising the melting temperature, but by improving maturation of GFP at high temperatures. I think this argument should definitely be provided in the paper, not just the response letter. Otherwise the reader is led to believe a 4 degree increase in the melting point is the mechanistic explanation for the phenotype, rather than the much more plausible effect on maturation.

Reviewer #2 (Remarks to the Author):

The authors answered and corrected the points raised by myself and the other reviewer. They added two new co-authors and performed the suggested experiments, which improved the manuscript's quality. Importantly, they toned down the arguments on the rescue of endangered species and cited previous work as requested. A question mark remains for me on how much their findings can be generalized, but this will be up to the reader to judge.

I thus recommend the manuscript in its updated form for publication.

RESPONSE TO REVIEWERS' COMMENTS

Reviewer #1 (Remarks to the Author):

The authors added some convincing *in vitro* translation assays that I think significantly strengthen their conclusions.

My only remaining concern is that they give a sensible explanation about how stabilizing mutations probably exert their effects not by raising the melting temperature, but by improving maturation of GFP at high temperatures. I think this argument should definitely be provided in the paper, not just the response letter. Otherwise the reader is led to believe a 4 degree increase in the melting point is the mechanistic explanation for the phenotype, rather than the much more plausible effect on maturation.

Reply: Thank you for the suggestion. In response, we have added the following explanation on lines 187-192 of pages 1:

Even though the proteins we study are thermally highly stable ($T_m > 70^\circ\text{C}$), a stabilizing mutation that increases the melting temperature by a few degrees can increase the amount of soluble fluorescent proteins at 44°C more than 60-fold (Fig. 3a). The reason is that elevated temperature does not just cause inactivation of the mature protein (quantified by T_m) but it can also interfere with the post-translational maturation of GFP⁴⁷.

Response to reviewer #2

Reviewer #2 (Remarks to the Author):

The authors answered and corrected the points raised by myself and the other reviewer. They added two new co-authors and performed the suggested experiments, which improved the manuscript's quality. Importantly, they toned down the arguments on the rescue of endangered species and cited previous work

as requested. A question mark remains for me on how much their findings can be generalized, but this will be up to the reader to judge.

I thus recommend the manuscript in its updated form for publication.

Reply: Thank you! We appreciate your helpful comments and suggestions for improving the quality of the manuscript!